# Categorical Functional Data Analysis. The cfda R Package

Cristian Preda [1,2,3,*] , Quentin Grimonprez [4] and Vincent Vandewalle [3,5]

1   UMR CNRS 8524—Laboratoire Paul Painlevé, University of Lille, 59000 Lille, France
2   Institute of Statistics and Applied Mathematics of the Romanian Academy, 050711 Bucharest, Romania
3   Inria Lille Nord-Europe, MODAL, 59655 Villeneuve d'Ascq, France
4   DiagRAMS Technologies, 59000 Lille, France; qgrimonprez@diagrams-technologies.com
5   Biostatistics Department, University of Lille, CHU Lille—ULR 2694 METRICS, 59000 Lille, France; vincent.vandewalle@univ-lille.fr
*   Correspondence: cristian.preda@univ-lille.fr

**Abstract:** Categorical functional data represented by paths of a stochastic jump process with continuous time and a finite set of states are considered. As an extension of the multiple correspondence analysis to an infinite set of variables, optimal encodings of states over time are approximated using an arbitrary finite basis of functions. This allows dimension reduction, optimal representation, and visualisation of data in lower dimensional spaces. The methodology is implemented in the cfda R package and is illustrated using a real data set in the clustering framework.

**Keywords:** functional data; categorical data; stochastic process; multiple correspondence analysis



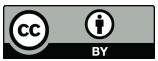

## 1. Introduction

Most literature devoted to functional data considers data as sample paths of a real-valued stochastic process, $X = \{X_t, t \in \mathcal{T}\}$, $X_t \in \mathbb{R}^p$, $p \geq 1$, where $\mathcal{T}$ is some continuous set. Among a considerable number of papers on the subject, the monographs of [1,2] are still the main references presenting methodologies for visualisation, denoising, clustering, and regression when dealing with functional data represented by real-valued functions. The fda R package [3,4] implements these methodologies and tools for such functional data.

In this paper, we consider the case where the underlying stochastic model generating the data is a continuous-time stochastic process $X = \{X_t, t \in \mathcal{T}\}$ such that for all $t \in \mathcal{T}$, $X_t$ is a categorical random variable rather than a real-valued one.

Let $(\Omega, \mathcal{A}, P)$ be a probability space, $\mathcal{S} = \{s_1, \ldots, s_K\}$, $K \geq 2$, be a set of $K$ states, and

$$X = \{X_t \; ; \; X_t : \Omega \longrightarrow \mathcal{S}, \quad t \in \mathcal{T}\} \tag{1}$$

be a family of categorical random variables indexed by $\mathcal{T}$. Thus, for some $\omega \in \Omega$, a path of $X$, $X(\omega)$, is a sequence of states $s_{i_j} = s_{i_j}(\omega)$ and time points $t_i = t_i(\omega)$ of transitions from one state to another one:

$$\{(t_0, s_{i_0}), (t_1, s_{i_1}), (t_2, s_{i_2}), \ldots\}, \tag{2}$$

where $0 = t_0 < t_1 < t_2 < \ldots$ are the jump times in $\mathcal{T}$, and $s_{i_j} \in \mathcal{S}$ with $i_j \in \{1, \ldots, K\}$, $\forall j \geq 0$. This path is read as follows. At time $t_0 = 0$, $\omega$ is in some state $s_{i_0}$; at time $t_1$, $t_1 > t_0$, $\omega$ moves randomly from $s_{i_0}$ to the state $s_{i_1}$; then, at time $t_2 > t_1$ it moves from the state $s_{i_1}$ to state $s_{i_2}$, and so on. If $\mathcal{T}$ is the interval of time $[0, T]$ for some $T > 0$, then the observation process stops when the time $T$ is reached or some absorbing state is observed.

We call the sample paths of $X$ given by sequences of type (2) *categorical functional data* generated by the process $X$.

Notice that there is no order assumption on the set of states. However, in some applications that order can be natural, all methodology developed in this paper being still valid. Figure 1 presents the graphical representation of one observation of a categorical functional random variable. The representation in (a) is appropriate when no natural order

relationship exists on the set of states $\mathcal{S}$, whereas the representation in (b) supposes that there exists some order relationship ($\prec$) on $\mathcal{S}$: $s_1 \prec s_2 \prec \ldots \prec s_K$.

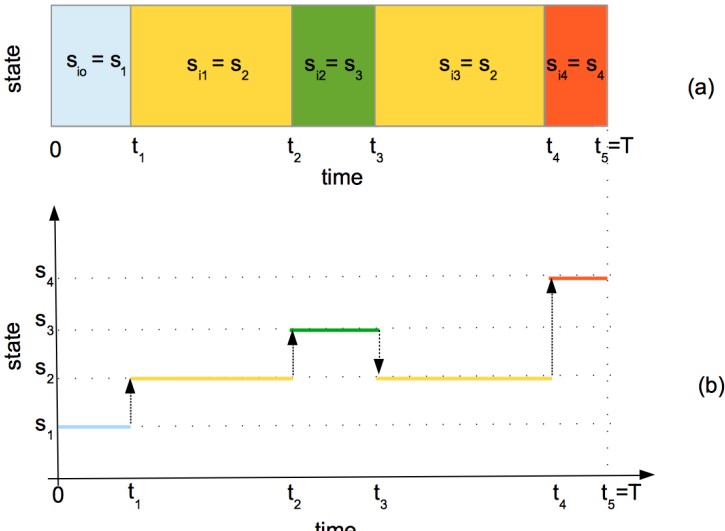

**Figure 1.** Categorical functional data: graphical representation of a path with (**a**) not ordered states, (**b**) ordered states.

To the best of our knowledge, and quite surprisingly, there is no recent research devoted to categorical functional data despite its ability to model real situations in different fields of applications: health and medicine (status of a patient over time) [5], economy (status of the market) [6,7], sociology (evolution of social status) [8,9], and so on. As a starting point on this topic, we mention the works of [7,8,10,11]. These works are devoted to the extension of factorial techniques (canonical and multiple correspondence analysis) towards functional data. The authors call that methodology "harmonic analysis". Applications of these techniques are presented in [9] for analysing career data and in [12] for studying the spectral properties of the transition probability matrix of a stationary Markovian jump process with continuous time. In [13], the authors cluster paths of semi-Markov processes using mixtures with application to sensory data. In [14], the authors present the harmonic analysis applied to the Brownian motion as an extension to an infinite (continuous) number of states.

In this article, we present categorical functional data analysis as an extension of the multiple correspondence analysis towards functional data and its implementation in the cfda R package. The theoretical foundations of this work are given in [8] and are based on the concept of *optimal encoding* of the states of the process $X$ with respect to maximum variance criterion among all encodings. In Section 2, we present the theoretical background of the optimal encoding methodology defining the *principal components* of the process $X$ throughout the optimal encodings. The approximation of the optimal encodings of the states into a basis of functions and optimal representation of categorical functional data in lower dimensional spaces are detailed. The implementation of the optimal encodings is presented throughout the cfda R package in Section 3, where an application on a real data set (care trajectories for patients diagnosed with severe infection) is performed in view of visualisation, descriptive statistics, and clustering.

## 2. Categorical Functional Data Analysis

Introduced in [8,11] under the name "analyse harmonique qualitative", multiple correspondence analysis is extended to categorical functional data. There are several ways to complete this; we have chosen in this work to introduce it as a problem of finding the latent variables (principal components) that are the most related to the process $X = \{X_t, t \in \mathcal{T}\}$.

Therefore, the principal components will enable defining the optimal encoding of states $\mathcal{S} = \{s_1, \dots, s_K\}$.

Without loss of generality, let us suppose that $\mathcal{T} = [0, T]$, with $T > 0$. For $x, y \in \mathcal{S}$, and $\forall t \in [0, T]$, let us denote by:

-   $\mathbf{1}_t^x = \begin{cases} 1 & \text{if } X_t = x, \\ 0 & \text{otherwise,} \end{cases}$

-   $p^x(t) = \mathsf{P}(X_t = x)$ and $p^{x,y}(t, s) = \mathsf{P}(X_t = x, X_s = y)$.

The general hypotheses considered in that framework are:

**Hypothesis 1.** **(H1)** *the process X is continuous in probability,*

$$\lim_{h \to 0} \mathsf{P}(X_{t+h} \neq X_t) = 0$$

and

**Hypothesis 2.** **(H2)** *for each time $t \in [0, T]$ (except possibly a finite discrete set of timepoints), any state has a strictly positive probability to occur:*

$$p^x(t) \neq 0, \forall x \in \mathcal{S}, \forall t \in [0, T].$$

*2.1. The Principal Components*

Let $L_2(\Omega)$ be the space of real random variables with finite second moment and, for some $t \in [0, T]$, $L(X_t)$ be the linear space spanned by $X_t$. Then, the conditional expectation operator associated to $X_t$,

$$\mathsf{E}_t : L_2(\Omega) \to L(X_t),$$

$$z \in L_2(\Omega), \quad z \longmapsto \mathsf{E}_t(z) = \sum_{x \in \mathcal{S}} \mathsf{E}(z|X_t = x)\mathbf{1}_t^x,$$

is also the orthogonal projector on the space linearly spanned by the set of indicator random variables $\{\mathbf{1}_t^x, x \in \mathcal{S}\}$. Notice that $\mathsf{E}_t$ is self-adjoint, idempotent, and of rank $K$.

For $z \in L_2(\Omega)$ and $t \in [0, T]$, the coefficient

$$\eta^2(z; X_t) = \frac{\mathsf{VAR}(\mathsf{E}_t(z))}{\mathsf{VAR}(z)}$$

is a measure of the correlation between $z$ and the variable $X_t$. The empirical version of $\eta^2$ is known as Wilks' Lambda statistics, well known in multivariate ANOVA [15].

Let us recall that if $t_1, t_2, \dots, t_p$ are $p$ different time points in $[0, T]$, then, the random variable $z$, which maximizes

$$\sum_{i=1}^{p} \eta^2(z; X_{t_i}) \tag{3}$$

defines the first principal component of the multiple correspondence analysis of the set of $p$ categorical variables $\{X_{t_1}, X_{t_2}, \dots X_{t_p}\}$ [16]. By an iterative procedure, the principal components of higher order are defined as maximizing (3) under orthogonality conditions with respect to the principal components of lower order.

In [8,11], the authors extend the multiple correspondence analysis to the process $X = \{X_t, t \in [0, T]\}$ (seen as an infinite set of categorical random variables). More specifically, the principal components are defined as the random variable $z$ that maximizes the criterion

$$\int_0^T \eta^2(z; X_t) dt. \tag{4}$$

They show that under the Hypotheses $H_1$ and $H_2$, the variable $z$, which maximizes (4) is the variable associated with the largest eigenvalue of the following (stochastic) eigenvalue problem:

$$\int_0^T \mathsf{E}_t(z)dt = \lambda z. \tag{5}$$

The operator $Q = \int_0^T \mathsf{E}_t dt$ is positive, hermitian, and compact. Therefore, $Q$ has a countable set of positive eigenvalues and eigenvectors, $\{(\lambda_i, z_i)\}_{i \geq 1}$ such that $\lambda_1 \geq \lambda_2 \geq \ldots \geq 0$ and

$$\int_0^T \mathsf{E}_t(z_i)dt = \lambda_i z_i.$$

The variables $\{z_i\}_{i \geq 1}$ are called *principal components* of the process $X$. Notice that $z = 1$ (constant) is an eigenvector of $Q$ associated to the largest eigenvalue $\lambda_{\max} = T$. It follows that the principal components $\{z_i\}_{i \geq 1}$ form a set of zero-mean uncorrelated random variables.

Moreover, we have that

$$\sum_{i \geq 1} \lambda_i = KT,$$

where $K$ is the number of states. Thus, excluding the trivial eigenvalue $\lambda_{\max} = T$, the contribution of the $i$-th principal component $z_i$ to (4) is

$$\mathrm{Ctr}(z_i) = \frac{\lambda_i}{(K-1)T}.$$

### 2.2. Optimal Encoding Functions

In order to solve (5), let denote by

$$\xi_t = \frac{1}{\lambda} \mathsf{E}_t(z), \quad \forall t \in [0, T]. \tag{6}$$

Under the Hypotheses $H_1$ and $H_2$, for each $t \in [0, T]$ $\xi_t$ is $X_t$-measurable, i.e., $\mathsf{E}_t \xi_t = \xi_t$ and $\{\xi_t\}_{t \in [0,T]}$ is a $L_2$-continuous stochastic process.

From (5) it follows that

$$z = \int_0^T \xi_t dt. \tag{7}$$

Taking the conditional expectation with respect to $X_t$ in (5), one obtains that the stochastic process $\{\xi_t\}_{t \in [0,T]}$ is eigenvector of the following (stochastic) eigenvalue problem posed in the space of $L_2$-continuous stochastic processes:

$$\int_0^T \mathcal{K}(t,s)\xi_s ds = \lambda \xi_t, \quad \forall t \in [0, T], \tag{8}$$

with $\mathcal{K}(t,s) = \mathsf{E}_t \mathsf{E}_s$, for all $t, s \in [0, T]$. Recall that the spectral analysis of the kernel $\mathcal{K}(t,s)$ yields to the canonical analysis of $X_t$ and $X_s$ [17].

It can be shown ([11]) that the eigenvalue problems (5) and (8) are equivalent in that sense that they have the same set of eigenvalues $\{\lambda_i\}_{i \geq 1}$ and there is an one-to-one correspondence between the principal components $z_i$ and the process $\xi_i = \{\xi_{i,t}, t \in [0, T]\}$, $\forall i \geq 1$. This correspondence is given by (6).

As in (5), the solution of (8) is unique up to a constant. To have unique eigenvectors, the usual constraint on $\{\xi_t\}_{t \in [0,T]}$ is that of total variance equals to one,

$$\int_0^T \mathsf{VAR}(\xi_t)dt = \int_0^T \mathsf{E}(\xi_t^2)dt = 1. \tag{9}$$

The relation (9) implies

$$\mathsf{VAR}(z) = \mathsf{E}(z^2) = \lambda. \tag{10}$$

It follows from (6) that $\xi_t$ is $X_t$-measurable for all $t \in [0, T]$, and one can write

$$\xi_t = \sum_{x \in \mathcal{S}} a^x(t)\mathbf{1}_t^x, \tag{11}$$

where $\{a^x\}_{x \in \mathcal{S}}$ are deterministic functions on $[0, T]$ that we call *optimal encoding* functions. Introducing (11) into (8) one obtains the following eigenvalue equation,

$$\int_0^T \sum_{y \in \mathcal{S}} p^{x,y}(t,s)a^y(s)ds = \lambda a^x(t)p^x(t), \quad \forall t \in [0, T], \forall x \in \mathcal{S}, \tag{12}$$

where $p^x(t) = \mathsf{P}(X_t = x)$, and $p^{x,y}(t,s) = \mathsf{P}(X_t = x, X_s = y)$.

The integral system (12) is a more "classic" one than (5) and (8). Under the Hypotheses $H_1$ and $H_2$, it admits the sequence of eigenvalues $\{\lambda_i\}_{i \geq 1}$ associated with the optimal encoding eigen-functions $\{a_i^x, x \in \mathcal{S}\}_{i \geq 1}$.

Notice that the constraint conditions in (9) are expressed now in terms of optimal encoding functions,

$$\int_0^T \sum_{x \in \mathcal{S}} [a^x(t)]^2 p^x(t)dt = 1. \tag{13}$$

According to (7), for $i \geq 1$, the $i$-th principal component $z_i$ is derived from the $i$-th optimal encoding functions $\{a_i^x\}$ as

$$z_i = \int_0^T \sum_{x \in \mathcal{S}} a_i^x(t)\mathbf{1}_t^x dt, \quad \forall i \geq 1. \tag{14}$$

### 2.3. Expansion Formulas and Dimension Reduction

As a summary of the previous section, the three equivalent eigen-problems stated in (5), (8), and (12) yield to the following elements of the analysis of $X$:

- the set of principal components $\{z_i\}_{i \geq 1}$ are zero-mean and uncorrelated:
  - $\mathsf{E}(z_i) = 0, \quad \forall i \geq 1.$
  - $\mathsf{COV}(z_i, z_j) = \begin{cases} \lambda_i & \text{if } i = j, \\ 0 & \text{otherwise.} \end{cases}$
- the set of eigen-processes $\{\xi_i = \{\xi_{i,t}, t \in [0, T]\}\}_{i \geq 1}$, which generates the principal components by (7),

$$z_i = \int_0^T \xi_{i,t}dt, \quad i \geq 1,$$

  are zero-mean and of unit total variance.
- the optimal encoding functions, $\{a_i^x = \{a_i^x(t), t \in [0, T]\}\}_{x \in \mathcal{S}, i \geq 1}$. They generate the eigen-processes $\xi_i$ by (11),

$$\xi_{i,t} = \sum_{x \in \mathcal{S}} a_i^x(t)\mathbf{1}_t^x, \quad \forall t \in [0, T].$$

  They satisfy the normalization condition (13).

**Expansion Formulas**

As an analogy to the Karhunen-Loève expansion for the scalar processes [18], the following expansion formulas hold [11]:

- for the process $X = \{X_t, t \in [0, T]\}$ throughout the indicators $\mathbf{1}^x = \{\mathbf{1}_t^x, t \in [0, T]\}$:

$$\mathbf{1}_t^x = \sum_{i \geq 1} z_i a_i^x(t) \frac{1}{p^x(t)}, \quad \forall x \in \mathcal{S}. \tag{15}$$

- for the bivariate joint probability function, $p^{x,y} = \{p^{x,y}(t,s), t,s \in [0,T]\}$: applying the Mercer theorem [19] to the integral Equation (12), one has the following expansion formula:

$$p^{x,y}(t,s) = p^x(t)p^y(s) \sum_{i \geq 1} \lambda_i a_i^x(t) a_i^y(s), \ \forall t,s \in [0,T], \forall x,y \in \mathcal{S}. \tag{16}$$

In particular, for $x = y$ and $s = t$, we obtain

$$p^x(t) = \left\{ \sum_{i \geq 1} \lambda_i [a_i^x(t)]^2 \right\}^{-1}, \ \forall t \in [0,T], \forall x \in \mathcal{S}. \tag{17}$$

**Dimension Reduction**

Using only the $q$ first terms in the right-side part of (15), $q \geq 1$, one obtains the best approximation of order $q$ of $X$ (viewed as a vector process $X = \{\mathbf{1}^x, x \in \mathcal{S}\}$) under the $L_2$ norm, among all the linear expansions of type

$$\mathbf{1}_t^x \approx \sum_{i=1}^q z_i a_i^x(t) \frac{1}{p^x(t)}, \ \forall x \in \mathcal{S}.$$

Thus, the $q$ first principal components,

$$\{z_1, \dots, z_q\}, \ q \geq 1,$$

allow for

- graphical representation of sample paths of $X$ in $\mathbb{R}^q$ (especially for $q = 2$, one obtains a 2-D representation of categorical functional data);
- fit of clustering and regression models with $X$ as explanatory variables;
- outliers or unusual data detection: in the context of real-valued functional data, in [20], the authors propose transformations and algorithms based on the concept of depth function in order to detect outliers. Transforming a real-valued functional variable into a categorical functional one (interval discretisation) and then performing optimal encoding can be an alternative to that proposed in [20].

*2.4. Approximation of Optimal Encoding Functions: A Basis Expansion Approach*

The eigenvalue equation in (12) provides the optimal encoding functions. For a two-state process, [11] considers the birth-and-death process on $[0,1]$,

$$X_t = \begin{cases} 0, & \text{if } t < \theta, \\ 1, & \text{if } t \geq \theta, \end{cases} \tag{18}$$

where $\theta$ is a random variable uniformly distributed on $[0,1]$. The authors provide in this case explicit formulas for the eigenvalues $\{\lambda_i\}_{i \geq 1}$, the optimal encoding functions $\{a_i^x\}_{i \geq 1}$, $x \in \mathcal{S}$, and the principal components $\{z_i\}_{i \geq 1}$. In [12], the author considers the case of stationary Markovian continuous time processes with reversible distribution. In this case, the system in (12) reduces to a system of linear second-order differential equations with constant coefficients.

In general, the solution of (12) is obtained by approximation. In their seminal work [8], the authors propose to approximate the encoding functions $\{a_i^x\}_{i \geq 1}$, $x \in \mathcal{S}$, into a basis of functions of dimension $m$, $m \geq 1$. As in the classical framework of functional data ([1]), the choice of $m$ is a tradeoff between complexity computation and precision of the approximation. In our simulation study (Section 4), we show the influence of the choice of $m$ on the approximation of optimal encodings.

Let $\{\phi_1, \ldots, \phi_m\}$, $\phi_i : [0, T] \to \mathbb{R}$, $i = 1, \ldots, m$, be a basis of functions (Fourier, B-splines, monomial, etc.), and for each $x \in \mathcal{S}$ consider the approximation:

$$a^x(t) \approx \alpha_{(x,1)}\phi_1(t) + \alpha_{(x,2)}\phi_2(t) + \ldots + \alpha_{(x,m)}\phi_m(t), \quad \forall t \in [0, T], \tag{19}$$

where $\boldsymbol{\alpha}_x = \left(\alpha_{(x,1)}, \alpha_{(x,2)}, \ldots, \alpha_{(x,m)}\right)' \in \mathbb{R}^m$ is the column vector of the expansion coefficients of $a^x$ into the basis $\{\phi_1, \ldots, \phi_m\}$.

Plugging (19) into (12) and (13), one obtains the following classical eigen-problem:

$$G\boldsymbol{\alpha} = \lambda F\boldsymbol{\alpha}, \tag{20}$$

under the constraint

$$\boldsymbol{\alpha}'F\boldsymbol{\alpha} = 1, \tag{21}$$

where $\boldsymbol{\alpha} \in \mathbb{R}^{m \times K}$ is the column vector obtained by the concatenation of the vectors $\{\boldsymbol{\alpha}_x\}_{x \in \mathcal{S}}$, and $G$ and $F$ are square matrices of size $mK \times mK$ defined as follows:

- The matrix $G$ is the covariance matrix of the random variables $\{V_{(x,i)}, x \in \mathcal{S}, i \in 1, \ldots m\}$, defined as

$$V_{(x,i)} = \int_0^T \phi_i(t)\mathbf{1}_t^x dt, \quad \forall x \in \mathcal{S}, \tag{22}$$

$$G = \left\{ G_{(x,i),(y,j)} = \text{COV}\left(V_{(x,i)}, V_{(y,j)}\right), \quad x, y \in \mathcal{S}, i, j = 1, \ldots, m \right\}, \tag{23}$$

$$
G = \begin{array}{c} \\ \\ (x,1) \\ \\ (x,i) \\ \\ (x,m) \\ \\ \end{array}
\begin{array}{c}
\cdots \quad (y,1) \quad \cdots \qquad\quad (y,j) \qquad\quad \cdots \quad (y,m) \quad \cdots
\end{array}
\left(
\begin{array}{cccccccc}
\cdots & \cdot & \cdots & & \cdot & & \cdots & \cdot & \cdots \\
\vdots & \vdots & \vdots & & \vdots & & \vdots & \vdots & \vdots \\
\cdots & \cdot & \cdots & & \cdot & & \cdots & \cdot & \cdots \\
\vdots & \vdots & \vdots & & \vdots & & \vdots & \vdots & \vdots \\
\cdots & \cdot & \cdots & \text{COV}\left(V_{(x,i)}, V_{(y,j)}\right) & & \cdots & \cdot & \cdots \\
\vdots & \vdots & \vdots & & \vdots & & \vdots & \vdots & \vdots \\
\cdots & \cdot & \cdots & & \cdot & & \cdots & \cdot & \cdots \\
\vdots & \vdots & \vdots & & \vdots & & \vdots & \vdots & \vdots \\
\end{array}
\right).
$$

- The matrix $F$ is defined by

$$F = \left\{ F_{(x,i),(y,j)} = \mathsf{E}\left(U_{(x,i),(y,j)}\right), \quad x, y \in \mathcal{S}, i, j = 1, \ldots, m \right\}, \tag{24}$$

where $U_{(x,i),(y,j)}$ is the random variable

$$U_{(x,i),(y,j)} = \int_0^T \phi_i(t)\phi_j(t)\mathbf{1}_t^x\mathbf{1}_t^y dt = \begin{cases} \int_0^T \phi_i(t)\phi_j(t)\mathbf{1}_t^x dt & \text{if } x = y, \\ \\ 0 & \text{otherwise.} \end{cases} \tag{25}$$

Thus, $F$ is a block diagonal matrix, each block being a square matrix of size $m \times m$ corresponding to each $x$ in $\mathcal{S}$, $\{U_{(x,i),(x,j)}, i, j = 1, \ldots, m\}$.

**Example 1.** *Let us observe that if $0 = t_0 < t_1 < \ldots < t_m = T$ is a sequence of timepoints in $[0, T]$, and for $i = 1, \ldots, m$, one defines the basis (B-splines of order 1),*

$$\phi_i(t) = \begin{cases} 1 & \text{if } t \in [t_{i-1}, t_i], \\ \\ 0 & \text{otherwise,} \end{cases}$$

*then, the random variable $V_{(x,i)}$ represents the time spent in the state $x$ in the interval $[t_{i-1}, t_i]$. Since $V_{(x,i)} = U_{(x,i),(x,i)}$, then $F$ is a diagonal matrix with elements $\mathsf{E}(V_{(x,i)})$, $x \in \mathcal{S}$, $i = 1, \ldots, m$.*

### 2.5. Estimation

Notice that the random variables $V_{(x,i)}$ and $U_{(x,i),(y,j)}$, $x, y \in \mathcal{S}$, $i, j = 1, \ldots, m$ are computed from $X = \{X_t, t \in [0, T]\}$ throughout the basis of functions $\{\phi_1, \ldots, \phi_m\}$.
Thus, if $\{X_1, \ldots, X_n\}$ is a sample of $n$ paths of $X$ corresponding to a random sample $(\omega_1, \ldots, \omega_n) \in \Omega^n$, then the corresponding samples $V_{(x,i)}(\omega)$ and $U_{(x,i),(x,j)}(\omega)$, $\omega \in \{\omega_1, \ldots, \omega_n\}$, provide two classical data sets, V and U, as:

- the V data set with $n$ rows and $Km$ columns for the $V$'s random variables,

$$
V = 
\begin{array}{c|ccccccc}
\omega & V_{(s_1,1)} & \cdots & V_{(s_1,m)} & \cdots & V_{(x,i)} & \cdots & V_{(s_K,m)} \\
\hline
\omega_1 & V_{(s_1,1)}(\omega_1) & \cdots & V_{(s_1,m)}(\omega_1) & \cdots & V_{(x,i)}(\omega_1) & \cdots & V_{(s_K,m)}(\omega_1) \\
\vdots & \vdots & \vdots & \vdots & \vdots & \vdots & \vdots & \vdots \\
\omega_n & V_{(s_1,1)}(\omega_n) & \cdots & V_{(s_1,m)}(\omega_n) & \cdots & V_{(x,i)}(\omega_n) & \cdots & V_{(s_K,m)}(\omega_n)
\end{array}
$$

- and the U dataset with $n$ rows and $Km^2$ columns for the $U$'s random variables, respectively:

$$
U = 
\begin{array}{c|ccccccc}
\omega & \cdots & U_{(x,i),(x,1)} & \cdots & U_{(x,i),(x,m)} & \cdots & U_{(s_K,m),(s_K,m)} \\
\hline
\omega_1 & \cdots & U_{(x,i),(x,1)}(\omega_1) & \cdots & U_{(x,i),(x,m)}(\omega_1) & \cdots & U_{(s_K,m),(s_K,m)}(\omega_1) \\
\vdots & \vdots & \vdots & \vdots & \vdots & \vdots & \vdots \\
\omega_n & \cdots & U_{(x,i),(x,1)}(\omega_n) & \cdots & U_{(x,i),(x,m)}(\omega_n) & \cdots & U_{(s_K,m),(s_K,m)}(\omega_n)
\end{array}
$$

Therefore, the matrices $G$ and $F$ are estimated from the sample $\{X_1, \ldots, X_n\}$ by the matrices $\widehat{G}$ and $\widehat{F}$, the covariance matrix estimator of the random variables $V$'s, and the mean estimator of the random variables $U$'s. For each $i$ and $j$ in $\{1, \ldots, m\}$ and $x$ and $y$ in $\mathcal{S}$, one has:

$$
\widehat{G}_{(x,i),(y,j)} = \widehat{\mathrm{COV}}\left(V_{(x,i)}, V_{(y,j)}\right) = \frac{1}{n-1}\left(\sum_{h=1}^{n} V_{(x,i)}(\omega_h) V_{(y,j)}(\omega_h) - n\bar{V}_{(x,i)}\bar{V}_{(y,j)}\right)
$$

and

$$
\widehat{F}_{(x,i),(y,j)} = \begin{cases} \bar{U}_{(x,i),(y,j)} = \dfrac{1}{n}\sum_{h=1}^{n} U_{(x,i),(y,j)}(\omega_h) & \text{if } x = y, \\ 0 & \text{otherwise.} \end{cases}
$$

An estimate of $i$-th eigen vector of (20), $i \geq 1$, is the $i$-th eigenvector $\widehat{\alpha}_i$ of the eigen-Equation (26),

$$
\widehat{G}\widehat{\alpha} = \widehat{\lambda}\widehat{F}\widehat{\alpha}, \tag{26}
$$

under the constraint

$$
\widehat{\alpha}'\widehat{F}\widehat{\alpha} = 1. \tag{27}
$$

Notice that, from the construction of $F$ and $G$ matrices, the elements of $\widehat{\boldsymbol{\alpha}}_i$ are indexed by the couple (state and basis function) i.e., $(x, j)$ with $x \in \mathcal{S}$ and $j \in \{1, \ldots, m\}$:

$$\widehat{\boldsymbol{\alpha}}_i = (\underbrace{\widehat{\alpha}_{i,(s_1,1)}, \ldots, \widehat{\alpha}_{i,(s_1,j)}, \ldots \widehat{\alpha}_{i,(s_1,m)}}_{\text{state } s_1}, \ldots, \underbrace{\widehat{\alpha}_{i,(x,1)}, \ldots, \widehat{\alpha}_{i,(x,j)}, \ldots \widehat{\alpha}_{i,(x,m)}}_{\text{state } x}, \ldots, \underbrace{\widehat{\alpha}_{i,(s_K,1)}, \ldots, \widehat{\alpha}_{i,(s_K,j)}, \ldots \widehat{\alpha}_{i,(s_K,m)}}_{\text{state } s_K}).$$

Then, for each $x \in \mathcal{S}$, the $i$-th encoding eigen-function $a_i^x$ is estimated by

$$\widehat{a}_i^x = \sum_{j=1}^{m} \widehat{\alpha}_{i,(x,j)} \phi_j, \quad i \geq 1. \tag{28}$$

The estimates for the encoding functions enable computing the principal components $z_i$ for each unit $\omega$ in the sample $(\omega_1, \ldots, \omega_n)$, as

$$\widehat{z}_i(\omega) = \int_0^T \sum_{x \in \mathcal{S}} \widehat{a}_i^x(t) \mathbf{1}_t^x(\omega) dt = \sum_{x \in \mathcal{S}} \sum_{j=1}^{m} \widehat{\alpha}_{i,(x,j)} V_{(x,j)}(\omega), \quad i \geq 1. \tag{29}$$

Notice that the variance of $\widehat{z}_i$ equals the $i$-th eigenvalue $\widehat{\lambda}_i$ of (26),

$$\widehat{\mathrm{VAR}(\widehat{z}_i)} = \widehat{\lambda}_i, \quad i \geq 1. \tag{30}$$

Confidence bounds.

Bootstrapping from the V and U datasets, through (26), one obtains an estimate of the covariance matrix of $\widehat{\alpha}_i$ denoted with $\widehat{\Sigma}_i$. Therefore, for each $t \in [0, T]$, we have

$$\widehat{\mathrm{VAR}(a_i^x(t))} = \boldsymbol{\phi}(t)' \widehat{\Sigma}_{(i,x)} \boldsymbol{\phi}(t),$$

where $\boldsymbol{\phi}(t)$ is the column vector $\boldsymbol{\phi}(t) = (\phi_1(t), \ldots, \phi_m(t))'$, and $\widehat{\Sigma}_{(i,x)}$ is the covariance matrix of $\widehat{\alpha}_{i,x} = \left( \widehat{\alpha}_{i,(x,1)}, \ldots, \widehat{\alpha}_{i,(x,m)} \right)$. Notice that $\widehat{\Sigma}_{(i,x)}$ is a submatrix of $\widehat{\Sigma}_i$.

Then, for a confidence level $1 - u$, $u \in [0, 1]$, a confidence interval for $a_i^x(t)$ is obtained as

$$\mathtt{CI}^{1-u}(a_i^x(t)) = \widehat{a_i^x(t)} \pm \zeta_{1-\frac{u}{2}} \sqrt{\widehat{\mathrm{VAR}(a_i^x(t))}},$$

where $\zeta_{1-\frac{u}{2}}$ is the quantile of order $1 - \frac{u}{2}$ of the standard normal distribution.

**Computational details**

- The approximation of optimal encoding functions in a basis of functions is based on the computation of random variables $V_{(x,i)}$ and $U_{(x,i),(x,j)}$ defined in (22) and (25), respectively. The computation of integrals involved in the definition of these random variables uses the `inprod` function of the fda R package which, at its turn, calls the function `eval.fd`. For $n$ and $K$ fixed, this step is the most computational in terms of time resources, and it depends on the number of basis functions, $m$, considered for the approximation (19). As the computation is performed for every $\omega$ in $(\omega_1, \ldots, \omega_n)$, parallel computation is performed.
- The $F$ matrix (24) can be singular in some specific situations, namely when there exists an interval $I \subset [0, T]$ and some state $x$ such that $p^x(t) = 0, \forall t \in I$. In this case, the hypothesis $H_2$ is not satisfied. For $t \in I$, the operator $E_t$ is degenerated; however, the eigenvalue Equation (5) is still valid. From (12), the optimal encoding function $a^x$ is not defined for $t \in I$. From a computational point of view, if $\phi_i$ is some element of the basis functions $\{\phi_1, \ldots, \phi_m\}$ with support in $I$, then the random variables $V(x, i)$ and $U_{(x,i),(\cdot,\cdot)}$ are zero-constant and therefore, the row and column corresponding to $(x, i)$ in the $F$ and $G$ matrices are zero vectors. Thus, the element $\alpha_{(x,i)}$ of the expansion coefficients vector $\alpha_x$ is not defined. Dropping the rows and columns from $F$ and

*G* corresponding to $(x, i)$ enables solving the eigen-problem in (26). Notice that the constraints (27) are fulfilled.

## 3. The cfda Package through Examples

The cfda R package (available online: https://cran.r-project.org/package=cfda, accessed on 28 November 2021) provides functions to analyze categorical functional data enabling computation of basic statistics (such as transition tables or visualisation) and the optimal encoding functions. It uses the ggplot2 [21] package to display graphics and the parallel [4] and pbapply [22] packages for code parallelization.

Many other R packages analyzing categorical data indexed by time exist, but they do not take into account the functional feature of data (continuous-time) or do not provide optimal representation of data for statistical learning purposes. The TraMineR [23] R package provides functions to perform descriptive analysis, and distance functions between sequences are defined to perform clustering analysis. The WeightedCluster [24] package relies on TraMineR and implements an associated clustering method. The msm [5] package estimates from categorical functional data and covariates a continuous-time (hidden) Markov multi-state model. The R packages ClickClust [25] and clickstream [26] deal with categorical functional data as discrete Markov chains and propose clustering methods based on mixture models.

### 3.1. Data

3.1.1. An Example of Real Dataset: Paths of Patients with Severe Infection

The cfda package is illustrated with the `care` dataset [27]. It contains 2929 care trajectories for patients diagnosed with a severe infection. Each month from the diagnosis of the infection, the followup of each patient is recorded using one of the following four states:

- "D": the patient has not a medical followup,
- "C": the patient has a medical followup but no treatment,
- "T": the patient has a medical followup with a treatment, but the infection is not suppressed,
- "S": the patient has a medical followup with a treatment, and the infection is suppressed.

The dataset is loaded running:

```
R> data(care)
R> head(care, 10)

id time state
 3    0     D
 3    5     D
 9    0     D
 9    1     D
13    0     D
13    7     D
15    0     D
15    4     T
15    7     C
15    8     D
```

The dataset is in a specific format required by every function in this package. Data must be provided as a data.frame with three columns named `id`, `time`, and `state`. The `id` column contains the identifiers of the statistical units (e.g., patient IDs), the `time` column contains the different timepoints of state changes ,and the `state` column contains the state that occurs at the corresponding time. For example, in the dataset above, four patients with IDs 3, 9, 13, and 15 are shown. The patient with ID 15 has an initial state (D) at time

$t = 0$, he stays in this state until $t = 4$ months, at which time he moves in a new state (`T`), and so on.

Note that within each trajectory, the time values must be ordered. Concerning `id` and `state`, the used format is quite versatile: character, factor, or integer can be used.

### 3.1.2. Visualize a Dataset

The `summary_cfd` function gives an overview of the dataset by printing information such as the number of paths, the time range, or the number of states, etc. All this information is returned in a list.

```
R> summary_cfd(care)

Number of rows: 10017
Number of individuals: 2929
Time Range: 0--50
Same time start value for all ids: TRUE
Same time end value for all ids: FALSE
Number of states: 4
States:
   D, T, C, S
Number of individuals visiting each state:
   D    C    T    S
2905 1154 1014 1063
```

Notice that all paths have the same start time value but not the same end time value. This does not meet the constraints for performing a functional data analysis. A subsample of paths observed until some specified time must be considered in order to perform optimal encoding computation.

A sample of paths from the `care` dataset is plotted using the `plotData` function (see Figure 2). Each line corresponds to a patient path in the dataset, the successive changes of states are represented by different colors.

```
plotData(data, col, addId, addBorder, sort)
```

The `plotData` function takes in argument a formatted data.frame (`data`) and additional aesthetic parameters:

`group`　a vector of the length of the number of paths of `data` containing a variable describing a group structure (if any) of data. Paths from different groups are displayed on different subplots. Paths whose group is coded as `NA` are ignored.

`addId`　a boolean to add the id of paths on the y-axis.

`addBorder`　a boolean to add the black border around each state.

`sort`　a boolean to sort paths according to the duration of their first state.

`col`　allows users to customize state colors by providing a vector of the same length as the number of state. `col` is a character (named) vector containing defined color names from R (e.g., c(''red'', ''blue'', ''darkgreen'')) or RGB colors (e.g., c(''#E41A1C'', ''#377EB8'', ''#4DAF4A'')).

`nCol`　only when `group` is used, the number of columns used to display different groups.

```
R> plotData(care[care$id <= 100, ])
R> plotData(care, addBorder = FALSE, addId = FALSE, sort = TRUE)
```

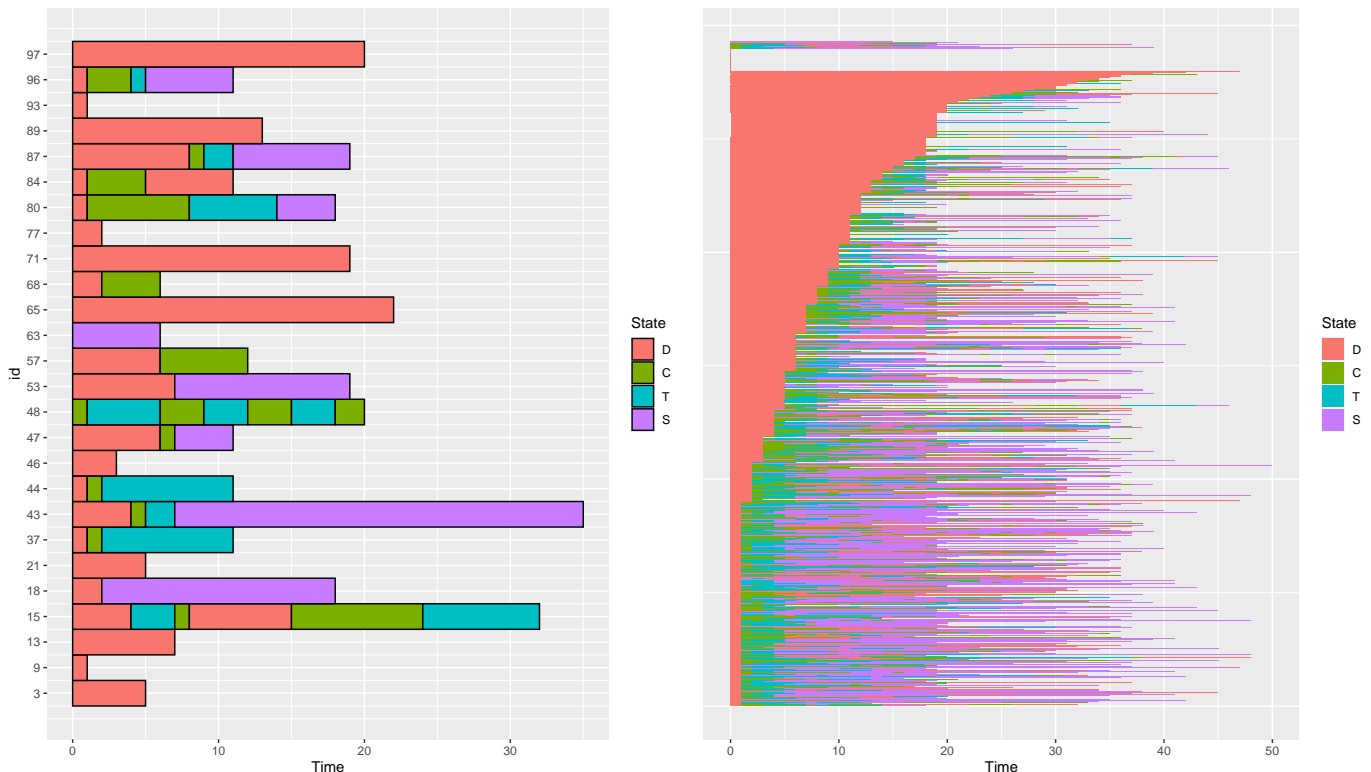

**Figure 2.** A sample (**left**) and all paths (**right**) from the `care` dataset plotted using the `plotData` function.

### 3.1.3. Paths of Same Length *T*

To compute the encoding functions, all paths must have the same start and end time. This is not the case in the `care` dataset. So, we need to select patients followed on a time interval of the same length, say $[0, T]$.

First, we compute the length (duration of followup) of each patient path using the `compute_duration` function. It returns a named vector, with the id as names, containing the duration. The results can be plotted using the `hist` function with the output of `compute_duration` as argument; it returns a `ggplot` object that can be modified.

```
R> duration <- compute_duration(care)
R> head(duration)

 3   9  13  15  18  21
 5   1   7  32  18   5

R> hist(duration)
```

The resulting plot is displayed in Figure 3. Most paths last less than 40 months with a mode value around 20 months. We decide to keep in the analysis a followup on the common interval $[0, 18]$.

To restrict paths to the interval $[0, 18]$, we use the `cut_data` function that has two parameters: `data` and `Tmax`, the maximal time value. After applying this function, the result contains all paths observed on $[0, \text{Tmax}]$.

```
R> idToKeep <- names(duration[duration >= 18])
R> care2 <- cut_data(care[care$id %in% idToKeep, ], 18)
R> head(care2)

   id time state
1  15    0     D
```

```
2  15    4      T
3  15    7      C
4  15    8      D
5  15   15      C
6  15   18      C
7  18    0      D
8  18    2      S
9  18   18      S
10 43    0      D
```

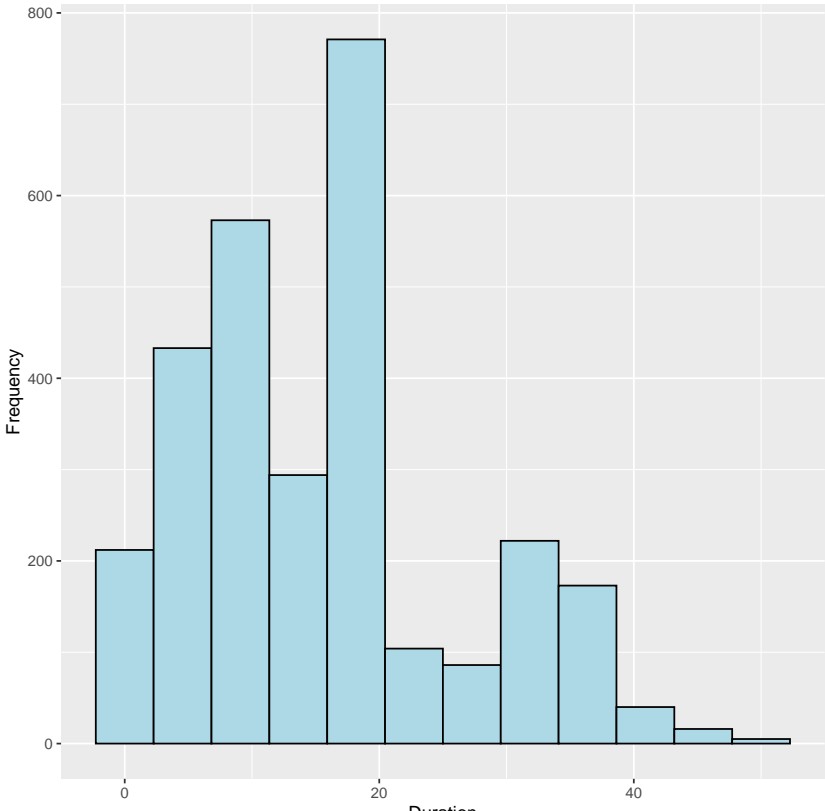

**Figure 3.** Distribution of the duration of trajectories.

### 3.2. Basic Statistics for Categorical Functional Data

3.2.1. Time Spent in Each State

An interesting statistic is the time spent in each state per patient that can be computed with `compute_time_spent` function. It returns a matrix with n rows (number of patients) and K columns (number of states) with the computed time. A `plot` function is provided to plot the distribution for each state. Figure 4 displays the graphic for the `care` dataset. We note that patients tend to stay longer without medical followup (D) than in the other states.

```
R> timeSpent <- compute_time_spent(care2)
R> head(timeSpent)

    D  C  T  S
15 11  4  3  0
18  2  0  0 16
43  4  1  2 11
48  0  7 11  0
53  7  0  0 11
65 18  0  0  0
```

```
R> boxplot(timeSpent)
```

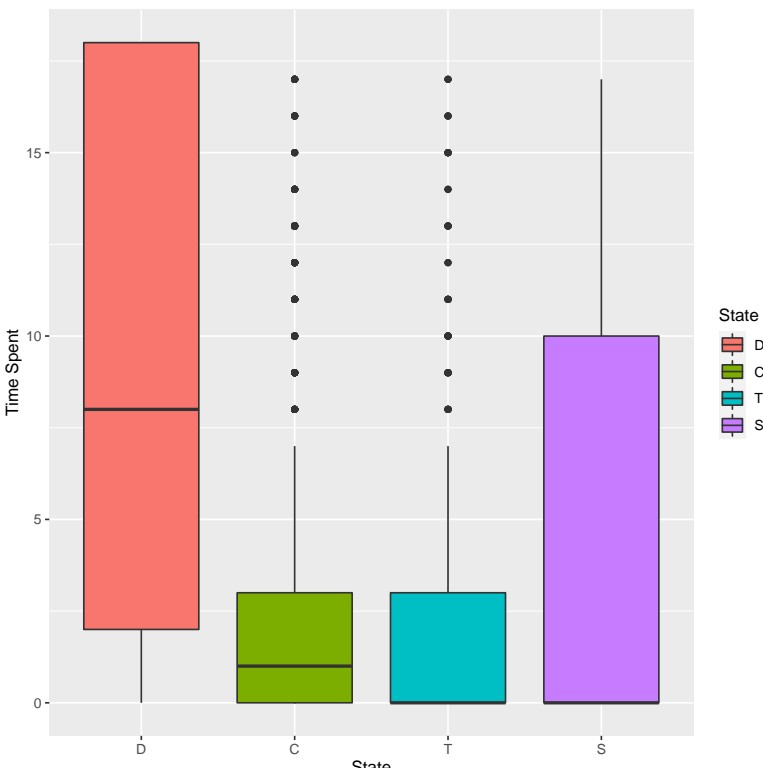

**Figure 4.** Distribution of time spent per state.

### 3.2.2. Number of Jumps

The `compute_number_jumps` function counts the number of transitions within paths. It has two arguments: `data`, the dataset in the right format and `countDuplicated`, a binary value indicating if jumps in the same state must be ignored (`FALSE`) or not, the default is `FALSE`. A `hist` function is provided to plot the distribution of the number of jumps. For the care dataset, the number of jumps varies between 0 and 8 (cf. Figure 5), with most patients with less than 6 jumps.

```
R> nJump <- compute_number_jumps(care2, countDuplicated = FALSE)
R> head(nJump)

15 18 43 48 53 65
 4  1  3  6  1  0

R> hist(nJump)
```

The transitions are visible using the `statetable` function that counts the number of transitions between each pair of states. Transitions between identical states can be removed from the output table using `removeDiagonal = TRUE`.

```
R> statetable(care2, removeDiagonal = TRUE)

     to
from   D   C   T   S
   D   0 697 253 146
   C 271   0 346  97
   T  16  74   0 461
   S  16  91  31   0
```

Notice that the state S (infection suppressed) is not an absorbing state, indicating some patients have relapsed.

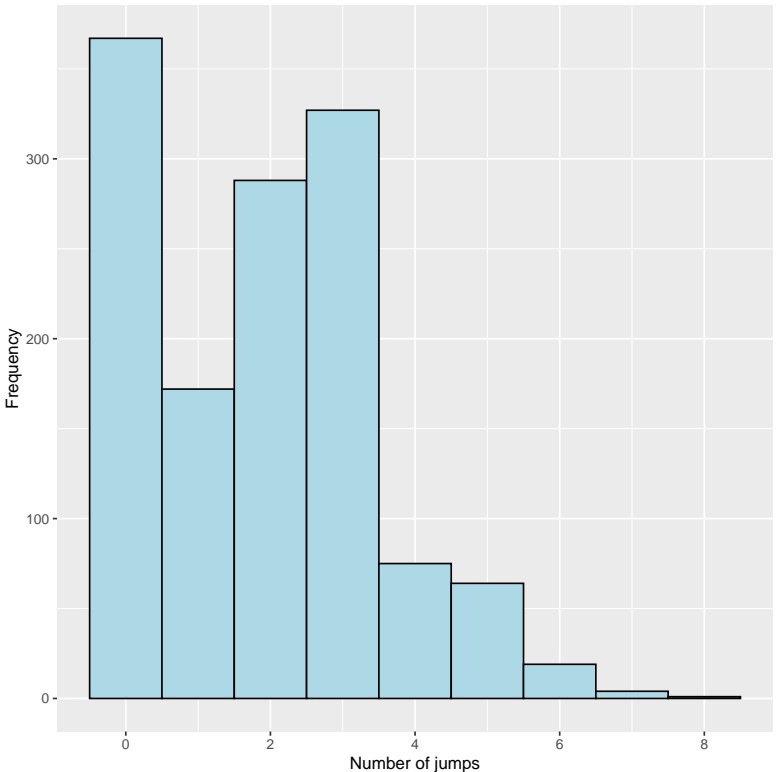

**Figure 5.** Distribution of number of jumps per individual.

### 3.2.3. States Distribution over Time

The last interesting statistic is the probability to be in some state at a given time. It is computed using the estimate_pt function. It has two arguments: data, the dataset in the right format and NAafterTmax. If NAafterTmax = FALSE, it considers that the last entry of an individual corresponds to its last change of state, i.e., the individual stays in the last recorded state for any time greater than the last time entry; if TRUE, it considers that for any time greater than the last time entry, the records are missing; the default is FALSE. This is an important parameter when paths have different ending times. This function returns a list of two elements: t, a vector containing the time values, and pt, a matrix (with the states in rows and the time values in columns) containing the computed probabilities. A plot function is provided to display the results; the first parameter is the output of estimate_pt function, the second is ribbon. If ribbon = FALSE, the probability for a state is displayed with a line, if TRUE, with a ribbon (cf. Figure 6). In this figure, we note that the probability of not having a followup (D) decreases over time, whereas the probability to be cured (S) has an opposite trend.

```
R> proba <- estimate_pt(care2)
R> proba

$pt
        0     1     2     3     4     5 ...
D 0.991 0.653 0.596 0.566 0.555 0.552 ...
C 0.008 0.202 0.180 0.166 0.156 0.134 ...
T 0.000 0.099 0.171 0.203 0.159 0.128 ...
S 0.001 0.046 0.053 0.065 0.131 0.185 ...
$t
 [1]  0  1  2  3  4  5  6 ...
```

```
R> plot(proba, ribbon = TRUE)
```

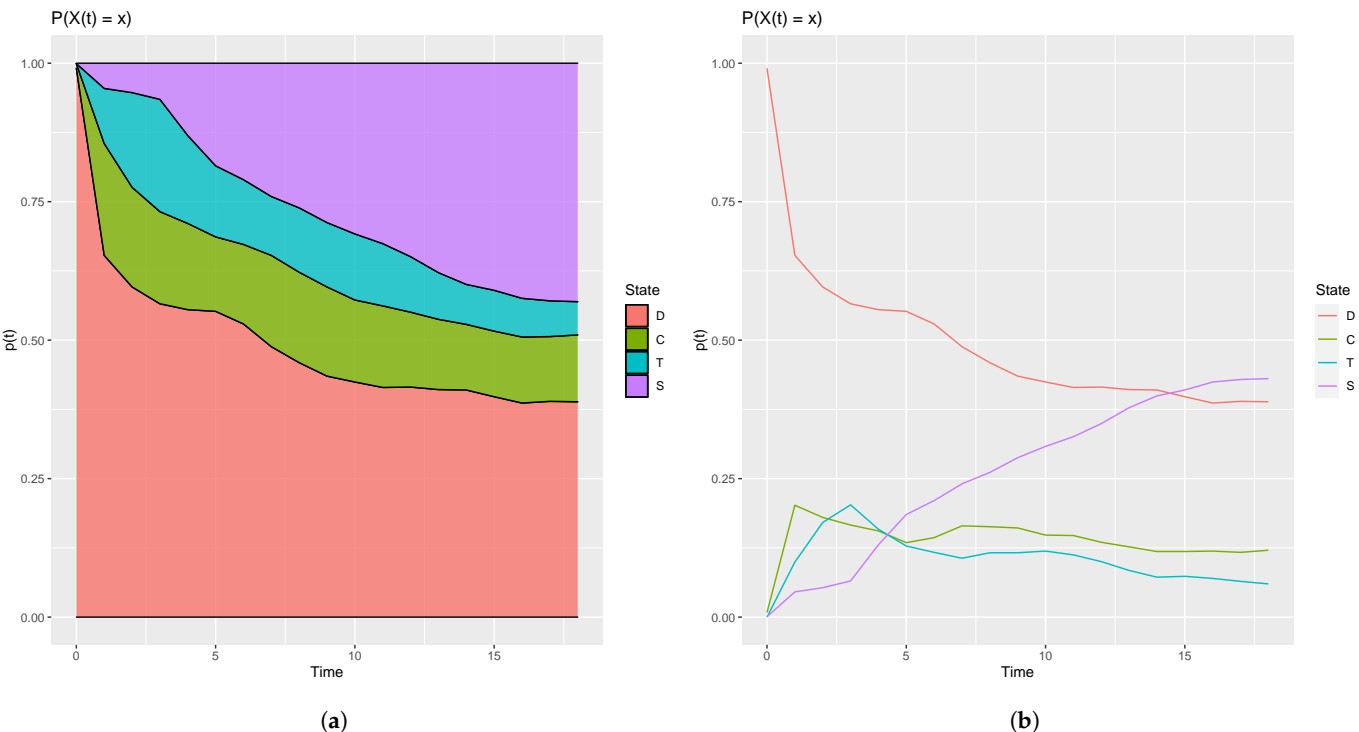

(**a**)  (**b**)

**Figure 6.** Probabilities to be in each state with regards to the time. (**a**) ribbon = TRUE; (**b**) ribbon = FALSE.

### 3.2.4. Continuous-Time Markov Chain

Let us assume data come from a continuous-time Markov process $X = \{X_t, t \in [0, T]\}$ with a set of states $\mathcal{S}$, $X_t \in \mathcal{S}, t \in [0, T]$. Then for $i, j \in \mathcal{S}$ and $t, s \geq 0$, the probability that the process will be in state $j$ at time $t + s$, given it is in state $i$ at time $s$, and the whole history until $s$ is given by:

$$\mathsf{P}(X_{s+t} = j | X_s = i, \{X_u = x_u : 0 \leq u < s\}) = \mathsf{P}(X_{s+t} = j | X_s = i).$$

A continuous-time Markov chain is completely described by its transition matrix $\Pi = (\pi_{ij})_{i,j \in \mathcal{S}}$ and $\lambda = \{\lambda_i\}_i \in \mathcal{S}$, the parameters of the exponentially distributed sojourn time in each state. See for more details [5].

The `estimate_Markov` function estimates the transition matrix ($\Pi$) and the $\lambda$ parameter (`lambda`) associated with the mean sojourn time spent in each state.

```
R> mark <- estimate_Markov(care2)
R> mark

$P
    to
from         D          C          T          S
   D 0.00000000 0.63594891 0.23083942 0.13321168
   C 0.37955182 0.00000000 0.48459384 0.13585434
   T 0.02903811 0.13430127 0.00000000 0.83666062
   S 0.11594203 0.65942029 0.22463768~0.00000000

$lambda
        D         C         T         S
0.1328033 0.2538578 0.2438443 0.1149503
attr(,``class'')
[1] ``Markov''
```

The estimated process can be plotted as a diagram with the `plot` function displayed in Figure 7. Each node represents a state with its mean sojourn time. An arrow between two nodes defines a possible transition with its probability.

```
R> plot(mark, main = ''care: transition graph'')
```

**care: transition graph**

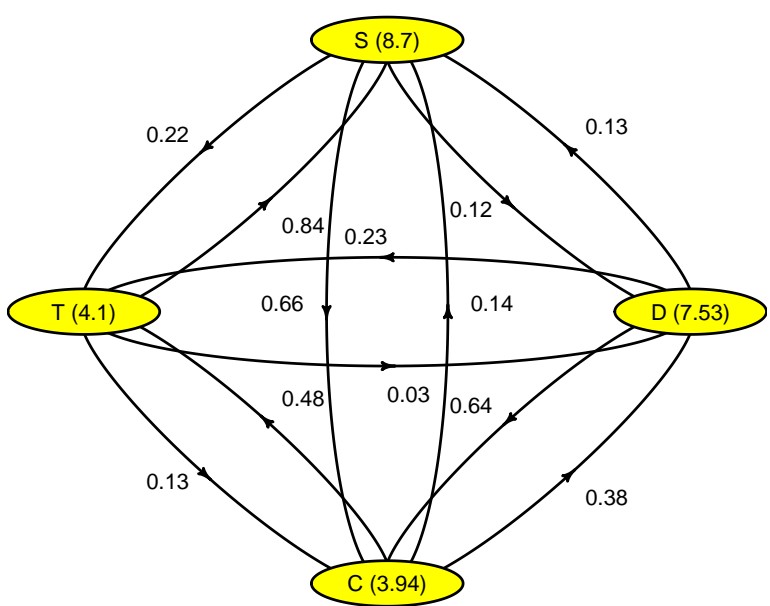

**Figure 7.** Transition graph displayed using `plot.Markov`.

*3.3. Optimal Encoding*

The main contribution of cfda is the computation of an optimal encoding for categorical functional data performed by the `compute_optimal_encoding` function. The two main parameters are `data`, the dataset in the cfda format, and `basisobj`, a `basisfd` object created using the different `create.*.basis` functions from the fda package. It also performs bootstrapping for computing confidence intervals of the computed encoding functions; associated parameters are `computeCI`, a logical indicating whether bootstrap must be performed, `nBootstrap`, the number of bootstrap samples, and `propBootstrap`, the proportion of individuals used for each bootstrap sample. Other parameters are `nCores` the number of cores to use, `verbose`, if `TRUE`, some information is printed during the process. The `compute_optimal_encoding` function uses `integrate` [4] to compute integrals, parameters for this function can be passed through ... in particular `subdivisions`, the number of subdivisions to estimate the integral.

```
R> set.seed(42)
R> basis <- create.bspline.basis(c(0, 18), nbasis = 10, norder = 4)
R> fmca <- compute_optimal_encoding(care2, basis, nCores = 7)

######### Compute encoding #########
Number of individuals: 1317
Number of states: 4
Basis type: bspline
Number of basis functions: 10
Number of cores: 7
---- Compute V matrix:
  |=================================================| 100% elapsed=21 s
```

```
DONE in 21.78 s
---- Compute U matrix:
  |================================================| 100% elapsed=122 s

DONE in 122.42 s
---- Compute encoding:
DONE in 0.13 s
---- Compute Bootstrap Encoding:
****************************************************
DONE in 1.3 s
Run Time: 149.84 s
```

The main part of the computational time comes from the computation of *V* and *U* matrices using parallel computation. Once these matrices are computed, a bootstrap estimation is performed for a low computational cost. The output object of `compute_optimal_encoding` is a list containing:

**eigenvalues** eigenvalues of the problem (26)

**alpha** coefficients of the different encoding for each eigenvector (a list of matrices) (26)

**pc** principal components for each eigenvector

**F** *F* matrix (see Equation (24))

**V** V matrix (see Equation (22))

**G** covariance matrix of V (see Equation (23))

**basisobj** `basisobj` parameter

**bootstrap** encoding for each bootstrap sample

**varAlpha** a list containing

$\widehat{\Sigma}_{(i,x)} \; \forall i, \forall x \in \mathcal{S}$, covariance matrix of $\widehat{\alpha}_{i,x} = \left( \widehat{\alpha}_{i,(x,1)}, \dots, \widehat{\alpha}_{i,(x,m)} \right)$

This object has its own `summary` and `print` functions.

3.3.1. Plot Functions

Three plot functions are associated with the `compute_optimal_encoding` function; the first argument of these functions is the output of `compute_optimal_encoding`.

The first one, the `plot` function plots the encodings associated with a given eigenvector (`harm` parameter, by default, the encodings associated with the first eigenvector are plotted). If `compute_optimal_encoding` was run with parameter `computeCI = TRUE`, then the confidence interval can be added on the plot using the parameter `addCI = TRUE`. A subset of states can be plotted by providing a vector with the state names to the `states` parameter.

The `plotEigenvalues` function plots the computed eigenvalues. It has two extra boolean parameters: `cumulative`, if TRUE, the cumulative sum of the eigenvalues is plotted and `normalize`, if TRUE, eigenvalues are normalized such that their sum is equal to 1.

The third one is the `plotComponent` function that plots paths coordinates using the principal components (`comp` parameter, a vector of length 2 containing the components number). The other arguments are `addNames` that adds the path's names on the plot and some parameters to adjust the position and size of these names (`nudge_x`, `nudge_y` and `size`).

The plots for the `care` dataset are shown in Figures 8 and 9 and are produced by the following code:

```
R> plotEigenvalues(fmca, cumulative = TRUE, normalize = TRUE)
R> plotComponent(fmca, comp = c(1, 2), addNames = FALSE)
```

```
R> plot(fmca)
R> plot(fmca, addCI = TRUE)
```

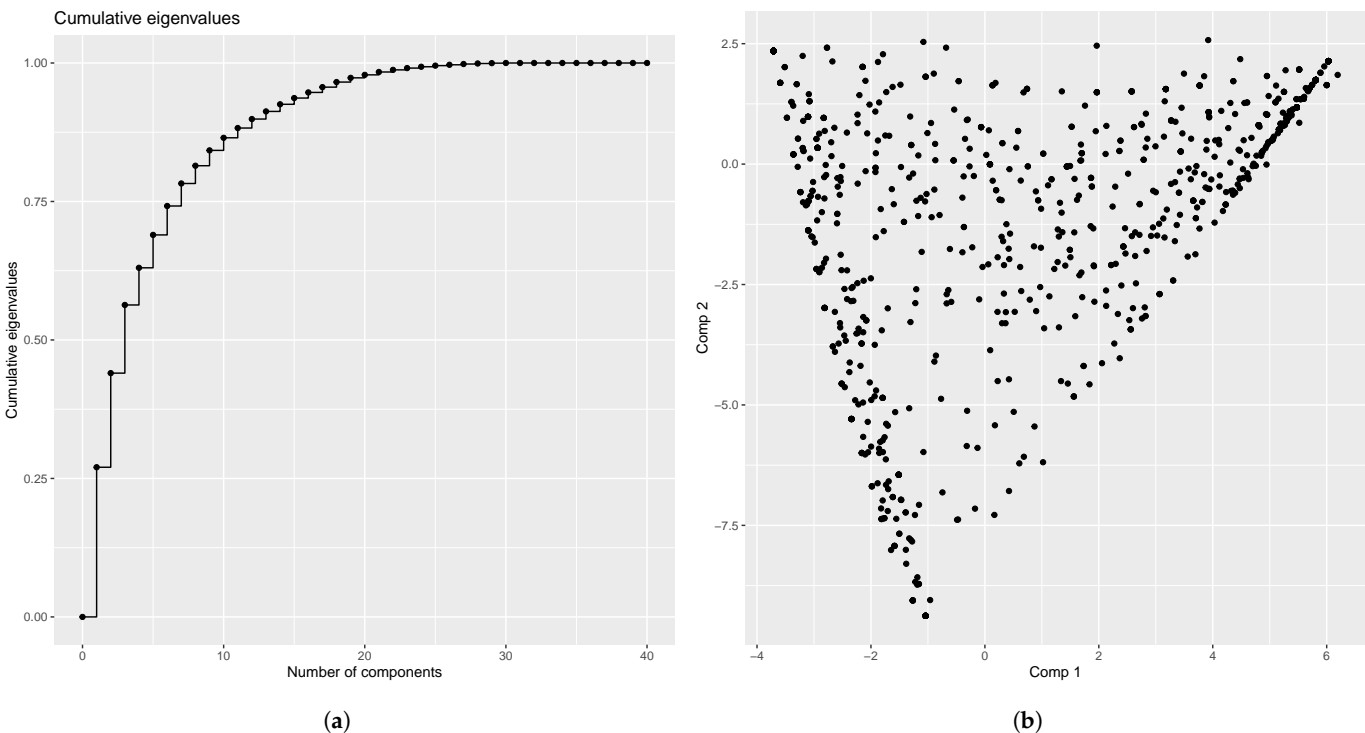

(**a**)                                                    (**b**)

**Figure 8.** Plots generated by different graphical functions on the output of the `compute_optimal_encoding` function. (**a**) `plotEigenvalues`; (**b**) `plotComponent`.

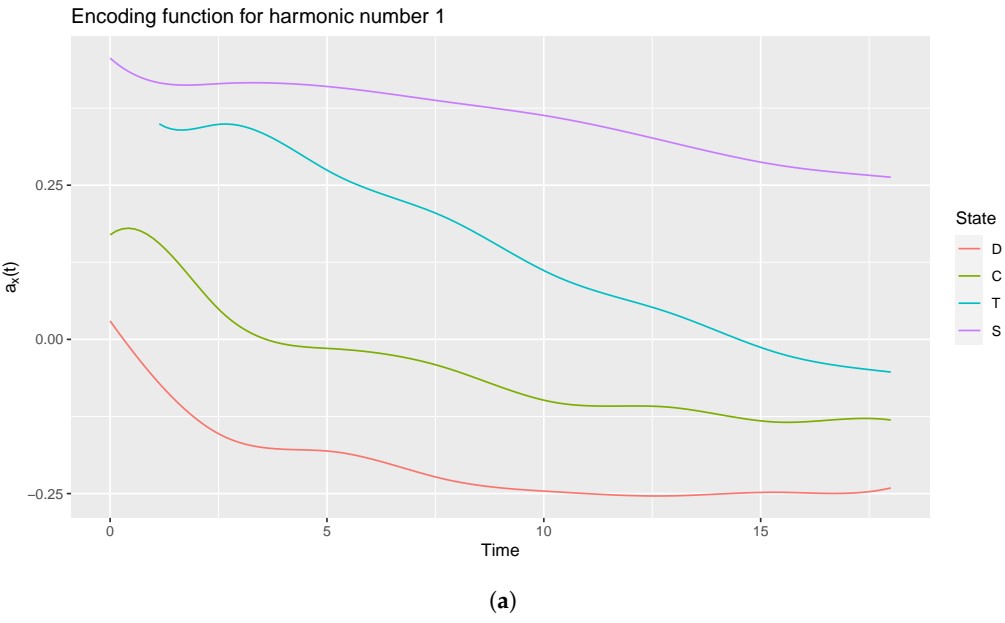

(**a**)

**Figure 9.** *Cont.*

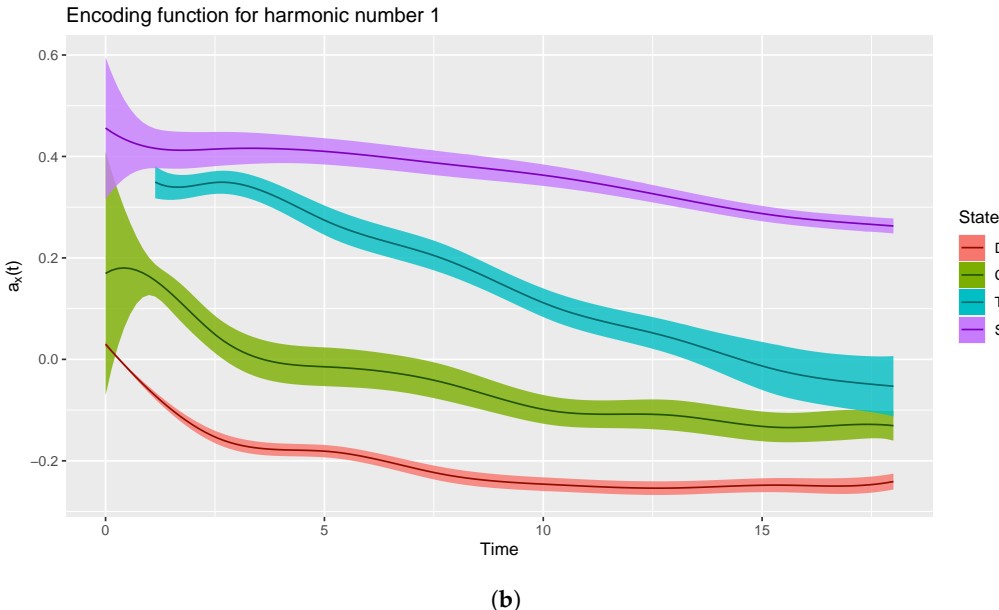

(**b**)

**Figure 9.** Plots generated by the `plot` function on the output of the `compute_optimal_encoding` function. (**a**) `plot(fmca)`; (**b**) `plot(fmca, addCI = TRUE)`.

### 3.3.2. Extract the Encoding Functions

The computed encoding functions can be extracted using the `get_encoding` function as an object of class `fd` (functional object from fda) using `fdObject = TRUE` or as a matrix using `fdObject = FALSE`. In the latter case, an extra parameter nx specifies the number of time points to extract.

```
R> encodingFd <- get_encoding(fmca, fdObject = TRUE)
R> str(encodingFd)

List of 3
 $ coefs  : num [1:10, 1:4] 0.0299 -0.0543 -0.1965 -0.1645 -0.2371 ...
  ..- attr(*, ``dimnames'')=List of 2
  .. ..$ : NULL
  .. ..$ : chr [1:4] ``D'' ``C'' ``T'' ``S''
 $ basis  :List of 10
  ..$ call       : language basisfd(type = type, | __truncated__
  ..$ type       : chr ``bspline''
  ..$ rangeval   : num [1:2] 0 18
  ..$ nbasis     : num 10
  ..$ params     : num [1:6] 2.57 5.14 7.71 10.29 12.86 ...
  ..$ dropind    : NULL
  ..$ quadvals   : NULL
  ..$ values     : list()
  ..$ basisvalues: list()
  ..$ names: chr [1:10] ``bspl4.1'' ``bspl4.2'' ``bspl4.3'' ``bspl4.4'' ...
  ..- attr(*, ``class'')= chr ``basisfd''
 $ fdnames:List of 3
  ..$ args: chr ``time''
  ..$ reps: chr [1:4] ``reps 1'' ``reps 2'' ``reps 3'' ``reps 4''
  ..$ funs: chr ``values''
 - attr(*, ``class'')= chr ``fd''

R> encodingMat <- get_encoding(fmca, fdObject = FALSE, nx = 19)
R> encodingMat
```

```
$x
 [1]   0  1  2  3  4  5  6  7  8  9 10 11 12 13 14 15 16 17~18

$y
                  D              C             T            S
 [1,]   0.02986969   0.169492601   0.50380590  0.4559043
 [2,]  -0.06073672   0.163315622   0.35643506  0.4180230
 [3,]  -0.12970105   0.088328506   0.34225746  0.4126089
 [4,]  -0.16758420   0.020411074   0.34651825  0.4159033
 [5,]  -0.17812958  -0.007566828   0.31652186  0.4149973
 [6,]  -0.18096760  -0.014569119   0.27436391  0.4100582
 [7,]  -0.19348217  -0.020533054   0.24249695  0.4020949
 [8,]  -0.21335627  -0.032358522   0.21809774  0.3925044
 [9,]  -0.23081796  -0.051930578   0.18880144  0.3828197
[10,]  -0.24069960  -0.077085840   0.15014660  0.3734934
[11,]  -0.24597159  -0.098545473   0.11162958  0.3630747
[12,]  -0.25009513  -0.107580629   0.08297660  0.3500902
[13,]  -0.25321503  -0.107980977   0.06221397  0.3347482
[14,]  -0.25359641  -0.110631333   0.04073503  0.3182122
[15,]  -0.25084388  -0.121299335   0.01379572  0.3018281
[16,]  -0.24813344  -0.132111070  -0.01301242  0.2873926
[17,]  -0.24881502  -0.133707940  -0.03278505  0.2766485
[18,]  -0.24943891  -0.128922501  -0.04469775  0.2692082
[19,]  -0.24091335  -0.130703127  -0.05297104  0.2629170
```

### 3.3.3. Interpreting the Encoding Functions

First, look at the plot of the encoding functions associated with the first principal component (harmonic 1, cf. Figure 9a).

```
R> plot(fmca, harm = 1)
```

At each time, the curve corresponding to state ''D'' is the lowest one, this indicates that paths with a "large" negative value for principal component number 1 tend to spend more time in this state, whereas those with large positive values will visit less this state. Similarly, individuals with a large positive value tend to spend more time in the state ''S''. To check these statements, individuals with extreme values on the first component are plotted using the plotData function with the group parameter. A group variable is created with two different values: ''min'' for the individuals with the 5% lowest value and ''max'' for the individuals with 5% highest value.

```
R> minpc1 <- names(which(fmca$pc[,1] <= quantile(fmca$pc[,1], 0.05)))
R> maxpc1 <- names(which(fmca$pc[,1] >= quantile(fmca$pc[,1], 0.95)))
R> ids <- unique(care2$id)
R> group <- factor(rep(NA, length(ids)), levels = c(''min'', ''max''))
R> group[ids %in% minpc1] = ''min''
R> group[ids %in% maxpc1] = "max"
R> plotData(care2, group = group, addId = FALSE, addBorder = FALSE,
+          sort = TRUE) +
+ ggplot2::labs(title = ''Extreme individuals on component 1'')
```

The result is visible in Figure 10 and confirms our interpretation. Clearly, patients in the ''min'' group spend 18 months in the ''D'' state (without medical followup), whereas patients in the ''max'' group spend most of their time in the state ''S'' (infection suppressed).

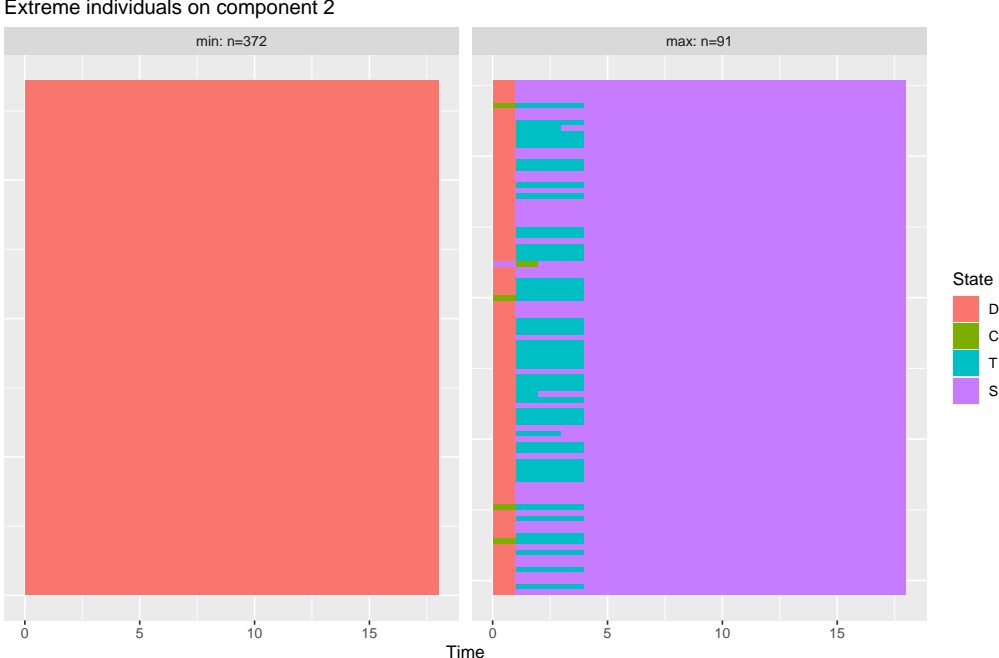

**Figure 10.** Individuals with extreme negative value (`min`) and extreme positive value (`max`) on the component 1.

### 3.3.4. Application to Clustering

The proposed method produces numerical encoding for categorical functional data. This encoding can be used for statistical learning purposes such as regression or clustering. In the following, we perform a hierarchical clustering to find hidden patterns (structure) in the `care` dataset.

The clustering is performed with the first principal components explaining at least 90% of the variance. The associated tree is displayed in Figure 11.

```
R> nPc90 <- which(cumsum(prop.table(fmca$eigenvalues)) > 0.9)[1]
R> hc <- hclust(dist(fmca$pc[, 1:nPc90]), method = ''ward.D2'')
R> plot(hc, labels = FALSE)
```

We decided to keep four clusters regarding the heights of the tree. The resulting clusters can be displayed using the `plotData` function with the `group` argument.

```
R> class <- cutree(hc, k = 4)
R> plotData(care2, group = class, addId = FALSE, addBorder = FALSE, +
+          sort = TRUE)
```

The different clusters are associated with the time spent in the different states after leaving the state ''D'' (cf. Figure 12). For example, the cluster number 1 corresponds to patients that have spent most of their time (after ''D'') in the ''C'' state.

A ready-to-run R code implementing optimal encoding for the care dataset towards clustering is presented in Appendix A.

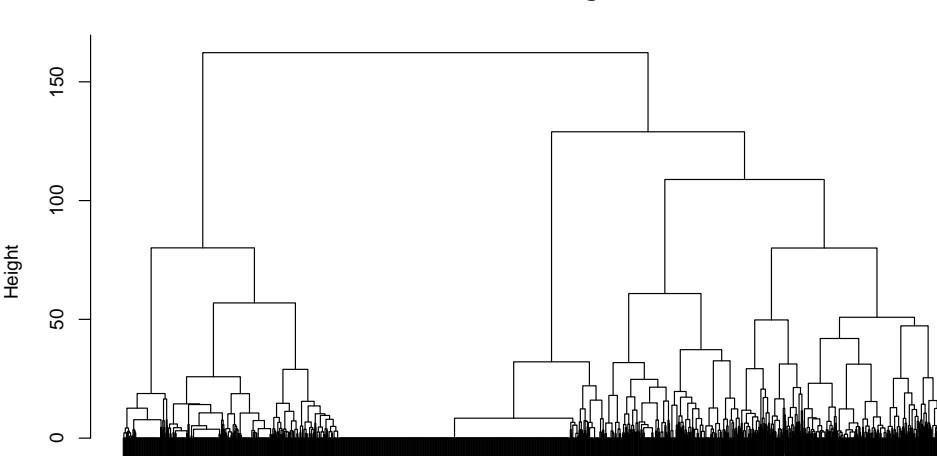

**Figure 11.** Hierarchical tree obtained using the principal components.

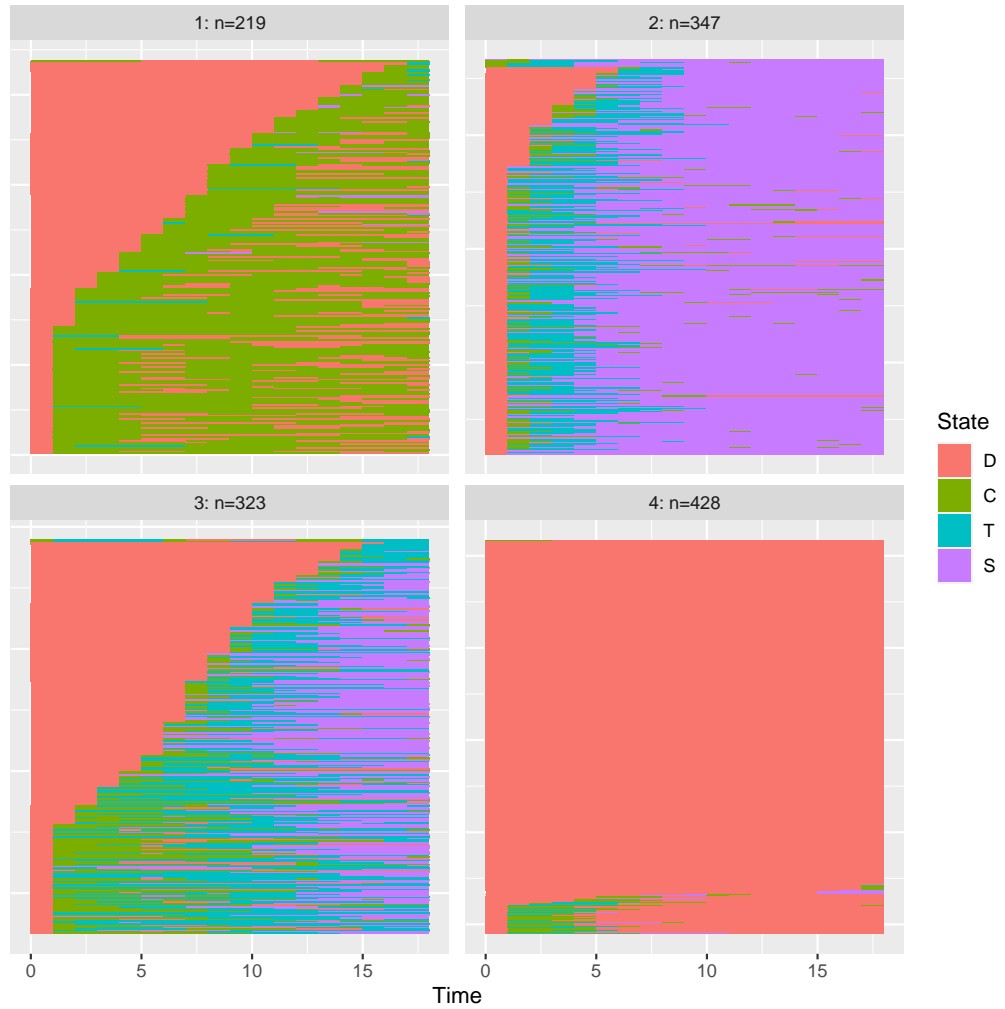

**Figure 12.** Content of the different clusters.

## 4. Simulation Study

*4.1. Birth-and-Death Process*

Data are simulated under the simple model of the birth-and-death process presented in [11]. The process is defined on the interval time $[0,1]$ by

$$X_t = \begin{cases} 0, & \text{if } t < \theta, \\ 1, & \text{if } t \geq \theta, \end{cases} \tag{31}$$

where $\theta$ is a random variable uniformly distributed on $[0,1]$.

In this case, $K = 2$, $\mathcal{S} = \{s_1 = 0, s_2 = 1\}$, and

$$p^x(t) = \mathsf{P}(X_t = x) = \begin{cases} t, & \text{if } x = 1 \\ 1 - t, & \text{if } x = 0. \end{cases}$$

For $t < s$, we have

$$p^{x,y}(t,s) = \mathsf{P}(X_t = x, X_s = y) = \begin{cases} 1 - s, & \text{if } x = 0, y = 0, \\ s - t, & \text{if } x = 0, y = 1, \\ 0 & \text{if } x = 1, y = 0, \\ t & \text{if } x = 1, y = 1, \end{cases}$$

and for $t > s$,

$$p^{x,y}(t,s) = \mathsf{P}(X_t = x, X_s = y) = \begin{cases} 1 - t, & \text{if } x = 0, y = 0, \\ 0 & \text{if } x = 0, y = 1, \\ t - s & \text{if } x = 1, y = 0, \\ s & \text{if } x = 1, y = 1. \end{cases}$$

From (12) and (14), the authors in [11] provide explicit formulas for the eigenvalues $\{\lambda_i\}_{i \geq 1}$, the optimal encoding functions $\{a_i^x\}_{i \geq 1}$, $x \in \mathcal{S}$, and the principal components $\{z_i\}_{i \geq 1}$, as follows:

- the eigenvalues are given by:

$$\lambda_i = \frac{1}{i(i+1)}, \quad i \geq 1. \tag{32}$$

- if $P_n(u) = \dfrac{1}{2^n n!} \dfrac{d^n}{du^n}(u^2 - 1)^n$ is the Legendre polynomials of order $n$, then the principal components $z_i$ corresponding to $\lambda_i$, are given, up to a constant, by:

$$z_i = P_i(2\theta - 1), \quad i \geq 1.$$

In particular, for $i = 1, 2$,

$$z_1 = \sqrt{6}\left(\theta - \frac{1}{2}\right)$$

is uniformly distributed on $\left[-\sqrt{\frac{3}{2}}; \sqrt{\frac{3}{2}}\right]$, and

$$z_2 = \sqrt{30}\left(\theta^2 - \theta + \frac{1}{6}\right).$$

Observe that $z_1$ and $z_2$ are linearly uncorrelated but related by

$$z_2 = \sqrt{\frac{5}{6}}\left(z_1^2 - \frac{1}{2}\right), \tag{33}$$

showing some regularity of the 2-D representation of data throughout the plot $\{(z_1(\omega), z_2(\omega), \omega \in \Omega\}$.

- for $i = 1, 2$, the optimal encoding functions are given by:

$$a_1^x(t) = \begin{cases} \sqrt{6}t, & \text{if } x = 0, \\[2mm] \sqrt{6}(t-1), & \text{if } x = 1, \end{cases} \tag{34}$$

and

$$a_2^x(t) = \begin{cases} \sqrt{120}\left(t^2 - \dfrac{t}{2}\right), & \text{if } x = 0, \\[3mm] \sqrt{120}\left(t^2 - \dfrac{3}{2}t + \dfrac{1}{2}\right), & \text{if } x = 1. \end{cases} \tag{35}$$

### 4.2. Results

We simulate data from the above process with different numbers of trajectories (individuals), $n = 50, 100, 200, 500, 1000$ and a B-spline basis function of order 4 with different numbers of basis functions $m = 5, 10, 20$ (equidistant knots). Simulation results are compared to the theoretical results presented in Section 4.1 to ensure the good behaviour of the implemented method.

#### 4.2.1. Eigenvalues

The first five eigenvalues for the different settings are compared to the eigenvalues from (32) in Table 1. The estimations are presented together with the associated standard errors in order to measure the impact of the choice of the sample size ($n$) and the dimension of the basis ($m$).

**Table 1.** True and estimated eigenvalues for the birth-and-death process. The estimated values are the mean over 100 samples. In brackets, the standard error is displayed.

| | | | | $m = 5$ | | |
|---|---|---|---|---|---|---|
| | **true** | $n = 50$ | $n = 100$ | $n = 200$ | $n = 500$ | $n = 1000$ |
| 1 | 0.5000 | 0.5117 ($6.4 \times 10^{-4}$) | 0.5013 ($4.4 \times 10^{-4}$) | 0.5000 ($3.1 \times 10^{-4}$) | 0.5009 ($1.8 \times 10^{-4}$) | 0.5018 ($1.5 \times 10^{-4}$) |
| 2 | 0.1667 | 0.1680 ($3.3 \times 10^{-4}$) | 0.1672 ($2.0 \times 10^{-4}$) | 0.1679 ($1.5 \times 10^{-4}$) | 0.1664 ($0.9 \times 10^{-4}$) | 0.1662 ($0.8 \times 10^{-4}$) |
| 3 | 0.0833 | 0.0824 ($1.9 \times 10^{-4}$) | 0.0835 ($1.4 \times 10^{-4}$) | 0.0834 ($0.9 \times 10^{-4}$) | 0.0835 ($0.5 \times 10^{-4}$) | 0.830 ($0.4 \times 10^{-4}$) |
| 4 | 0.0500 | 0.0455 ($1.3 \times 10^{-4}$) | 0.0490 ($0.9 \times 10^{-4}$) | 0.0494 ($0.6 \times 10^{-4}$) | 0.0492 ($0.4 \times 10^{-4}$) | 0.493 ($0.3 \times 10^{-4}$) |
| 5 | 0.0333 | 0.0184 ($0.6 \times 10^{-4}$) | 0.0205 ($0.4 \times 10^{-4}$) | 0.0211 ($0.3 \times 10^{-4}$) | 0.0215 ($0.2 \times 10^{-4}$) | 0.0216 ($0.1 \times 10^{-4}$) |
| | | | | $m = 10$ | | |
| | **true** | $n = 50$ | $n = 100$ | $n = 200$ | $n = 500$ | $n = 1000$ |
| 1 | 0.5000 | 0.5124 ($6.4 \times 10^{-4}$) | 0.5016 ($4.3 \times 10^{-4}$) | 0.5002 ($3.1 \times 10^{-4}$) | 0.5009 ($1.8 \times 10^{-4}$) | 0.5018 ($1.5 \times 10^{-4}$) |
| 2 | 0.1667 | 0.1692 ($3.4 \times 10^{-4}$) | 0.1677 ($2.0 \times 10^{-4}$) | 0.1682 ($1.5 \times 10^{-4}$) | 0.1665 ($0.9 \times 10^{-4}$) | 0.1663 ($0.8 \times 10^{-4}$) |
| 3 | 0.0833 | 0.0841 ($2.0 \times 10^{-4}$) | 0.0843 ($1.4 \times 10^{-4}$) | 0.0839 ($0.9 \times 10^{-4}$) | 0.0837 ($0.5 \times 10^{-4}$) | 0.831 ($0.4 \times 10^{-4}$) |
| 4 | 0.0500 | 0.0486 ($1.3 \times 10^{-4}$) | 0.0510 ($0.9 \times 10^{-4}$) | 0.0508 ($0.7 \times 10^{-4}$) | 0.0501 ($0.4 \times 10^{-4}$) | 0.0501 ($0.3 \times 10^{-4}$) |
| 5 | 0.0333 | 0.0317 ($0.8 \times 10^{-4}$) | 0.0335 ($0.6 \times 10^{-4}$) | 0.0334 ($0.4 \times 10^{-4}$) | 0.0336 ($0.2 \times 10^{-4}$) | 0.0335 ($0.2 \times 10^{-4}$) |
| | | | | $m = 20$ | | |
| | **true** | $n = 50$ | $n = 100$ | $n = 200$ | $n = 500$ | $n = 1000$ |
| 1 | 0.5000 | 0.5128 ($6.4 \times 10^{-4}$) | 0.5018 ($4.4 \times 10^{-4}$) | 0.5003 ($3.1 \times 10^{-4}$) | 0.5010 ($1.8 \times 10^{-4}$) | 0.5018 ($1.5 \times 10^{-4}$) |
| 2 | 0.1667 | 0.1699 ($3.4 \times 10^{-4}$) | 0.1681 ($2.0 \times 10^{-4}$) | 0.1683 ($1.5 \times 10^{-4}$) | 0.1666 ($0.9 \times 10^{-4}$) | 0.1663 ($0.8 \times 10^{-4}$) |
| 3 | 0.0833 | 0.0849 ($2.0 \times 10^{-4}$) | 0.0847 ($1.4 \times 10^{-4}$) | 0.0841 ($0.9 \times 10^{-4}$) | 0.0837 ($0.5 \times 10^{-4}$) | 0.0832 ($0.4 \times 10^{-4}$) |
| 4 | 0.0500 | 0.0495 ($1.3 \times 10^{-4}$) | 0.0514 ($0.9 \times 10^{-4}$) | 0.0510 ($0.6 \times 10^{-4}$) | 0.0502 ($0.4 \times 10^{-4}$) | 0.0501 ($0.3 \times 10^{-4}$) |
| 5 | 0.0333 | 0.0327 ($0.8 \times 10^{-4}$) | 0.0340 ($0.6 \times 10^{-4}$) | 0.0337 ($0.5 \times 10^{-4}$) | 0.0337 ($0.2 \times 10^{-4}$) | 0.0335 ($0.2 \times 10^{-4}$) |

### 4.2.2. Encoding Functions

Figure 13 shows the mean over 100 samples of the first and second encoding functions for the state 0 for $m = 5$. The true encoding functions (34) and (35) are displayed in solid black line. As for the eigenvalues, the best estimates are achieved with $n = 200$ and $n = 500$. The same conclusion holds for $m = 10$ and $m = 20$, but the number of basis functions does not seem to influence the accuracy.

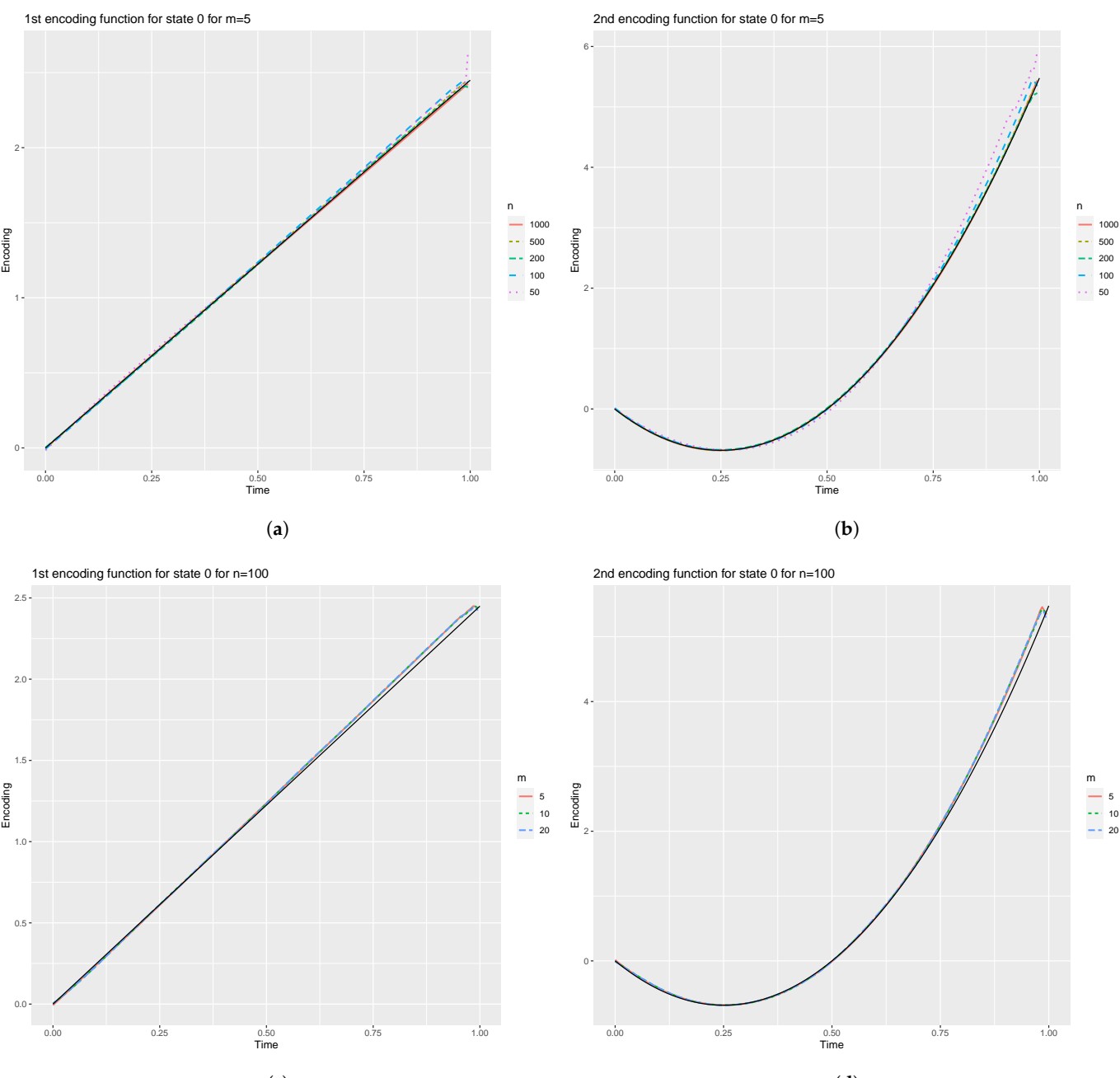

**Figure 13.** True (solid black) and estimated encoding functions for state 0 of the birth-and-death process. The estimated encoding functions are the mean of 100 samples. (**a**) First encoding function for state 0 ($m = 5$); (**b**) Second encoding function for state 0 ($m = 5$); (**c**) First encoding function for state 0 ($n = 100$); (**d**) Second encoding function for state 0 ($n = 100$).

### 4.2.3. Principal Components

In Figure 14, we check the relation between the first and second principal component (33). The theoretical equation is displayed in black, whereas the computed principal components

for a sample with $n = 500$ and $m = 20$ are in red. We note the closeness of the computed components with the theoretical equation.

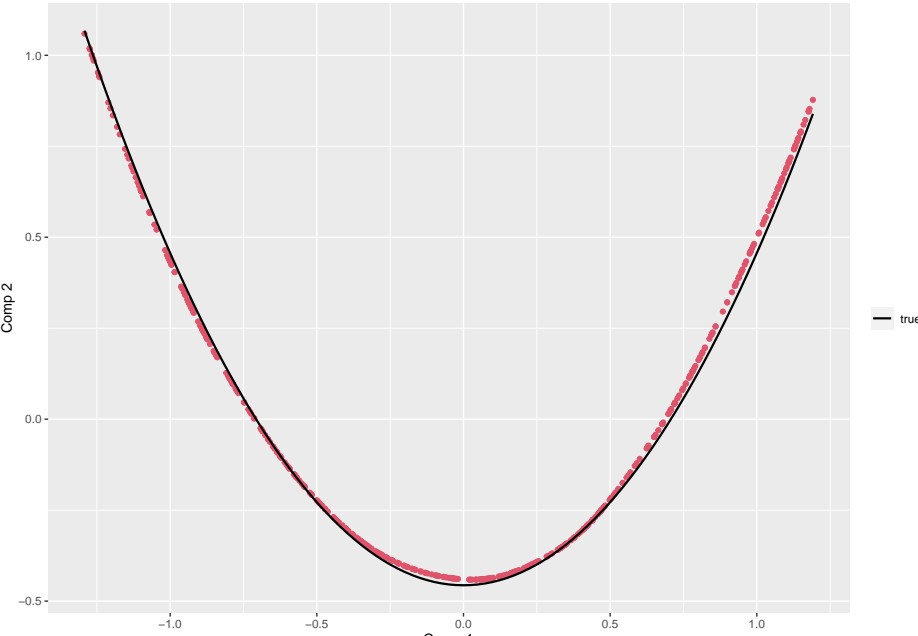

**Figure 14.** In red, first and second principal components for every observation of the birth-and-death process for $n = 500$ and $m = 20$. In solid black, the theoretical relation between these two components.

In Figure 15, the cumulative distribution functions (cdf) for the two first principal components are displayed as well as their empiric equivalent for $n = 500$ and $m = 20$. As described above, $z_1$ follows a uniform distribution between $-\sqrt{\frac{3}{2}}$ and $\sqrt{\frac{3}{2}}$, the empirical cdf (in red) is close to the theoretical one. The same representation is made for $z_2$.

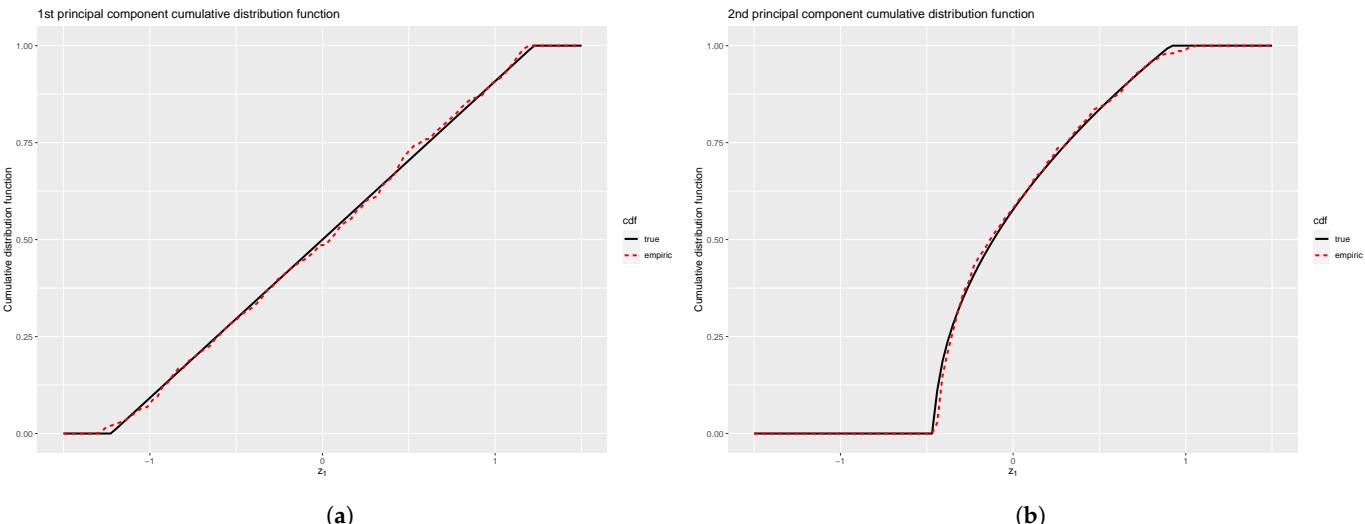

|     |     |
| :-: | :-: |
| (**a**) | (**b**) |

**Figure 15.** Empirical (resp. theoretical) cumulative distribution function for the first ($z_1$) and second ($z_2$) principal components of the birth-and-death process for $n = 500$ and $m = 20$. (**a**) Cumulative distribution function for $z_1$; (**b**) Cumulative distribution function for $z_2$.

The results in this paper were obtained using R 4.0.3 with the cfda 0.9.9 package. R itself and all packages used are available from the Comprehensive R Archive Network (CRAN) [4].

## 5. Summary and Discussion

Categorical functional data are represented by paths of a continuous-time stochastic process with values in a finite set of states. This is less popular than the real-valued functional data, which are yet another kind of infinite dimensional object. The analysis of categorical functional data is presented in this paper as an extension of the multiple correspondence analysis for the finite dimensional setting. Principal components, optimal encoding functions, and optimal representations are presented. A simulation study and a real data application illustrate the methodology implemented in the cfda R package.

In future work, we propose to address the problem of missing and noisy data. More specifically, timepoints are observed with noise or/and are missing.

**Author Contributions:** Conceptualization, C.P.; methodology, C.P., Q.G. and V.V.; software, Q.G. and C.P.; validation, C.P., Q.G. and V.V.; formal analysis, C.P.; investigation, C.P., Q.G. and V.V.; resources, Q.G., C.P. and V.V.; writing–original draft preparation, C.P., Q.G. and V.V.; writing–review and editing, C.P., Q.G. and V.V.; supervision, C.P.; project administration, C.P. and Q.G. All authors have read and agreed to the published version of the manuscript.

**Funding:** This research received no external funding.

**Institutional Review Board Statement:** Not applicable.

**Informed Consent Statement:** Not Applicable.

**Conflicts of Interest:** The authors declare no conflict of interest.

## Appendix A

**R Code: Care Application**

```
R> library(cfda)

R> # load the dataset
R> data(care)

R> summary_cfd(care)

R> # visualize the dataset
R> plotData(care[care$id <= 100, ])
R> plotData(care, addId = FALSE, addBorder = FALSE, sort = TRUE)

R> duration <- compute_duration(care)
R> head(duration)
R> hist(duration)

R> ####### Select individuals for encoding
R> ## We keep individuals with at least 18 months of follow-up and works on the first 18 months
R> length(duration[duration >= 18])

R> idToKeep <- as.numeric(names(duration[duration >= 18]))
R> care2 <- cut_data(care[care$id %in% idToKeep, ], 18)

R> head(care2, 10)

R> summary_cfd(care2)

R> plotData(care2, addId = FALSE, addBorder = FALSE, sort = TRUE)

R> ####### Basic statistics
```

```
R> timeSpent <- compute_time_spent(care2)
R> boxplot(timeSpent)

R> nJump <- compute_number_jumps(care2, countDuplicated = FALSE)
R> head(nJump)
R> hist(nJump)

R> statetable(care2, removeDiagonal = TRUE)

R> # individuals have not the same length and the last state is not necessarily an absorbing state,
   so we use NAafterTmax = TRUE
R> proba <- estimate_pt(care2, NAafterTmax = TRUE)
R> plot(proba, ribbon = TRUE)
R> plot(proba)

R> mark <- estimate_Markov(care2)
R> plot(mark, main = ``care: transition graph'')

R> ####### Encoding
R> set.seed(42)
R> basis <- create.bspline.basis(c(0, 18), nbasis = 10, norder = 4)
R> fmca <- compute_optimal_encoding(care2, basis, nCores = 7)

R> plotEigenvalues(fmca, cumulative = TRUE, normalize = TRUE)
R> plot(fmca)
R> plot(fmca, addCI = TRUE)
R> plotComponent(fmca, addNames = FALSE)

R> encodingFd <- get_encoding(fmca, fdObject = TRUE)
R> encodingMat <- get_encoding(fmca, fdObject = FALSE, nx = 19)

R> ## interpreting the results
R> plot(fmca, harm = 1)

R> minpc1 <- names(which(fmca$pc[,1] <= quantile(fmca$pc[,1], 0.05)))
R> maxpc1 <- names(which(fmca$pc[,1] >= quantile(fmca$pc[,1], 0.95)))

R> group <- rep(NA, length(unique(care2$id)))
R> group[unique(care2$id) %in% minpc1] = "min"
R> group[unique(care2$id) %in% maxpc1] = "max"

R> plotData(care2, group = group, addId = FALSE, addBorder = FALSE, sort = TRUE) +
R>   ggplot2::labs(title = ``Extreme individuals on component 1'')

R> plot(fmca, harm = 2)

R> minpc2 <- names(which(fmca$pc[,2] <= quantile(fmca$pc[,2], 0.05)))
R> maxpc2 <- names(which(fmca$pc[,2] >= quantile(fmca$pc[,2], 0.95)))

R> group <- rep(NA, length(unique(care2$id)))
R> group[unique(care2$id) %in% minpc1] = "min"
```

```
R> group[unique(care2$id) %in% maxpc1] = "max"

R> plotData(care2, group = group, addId = FALSE, addBorder = FALSE, sort = FALSE) +
R>   ggplot2::labs(title = ''Extreme individuals on component 2'')

R> ####### Clustering
R> nPc90 <- which(cumsum(prop.table(fmca$eigenvalues)) > 0.9)[1]
R> hc <- hclust(dist(fmca$pc[, 1:nPc90]), method = ''ward.D2'')

R> plot(hc, labels = FALSE)
R> barplot(rev(hc$height)[1:20])

R> cluster <- cutree(hc, k = 4)
R> plotData(care2, group = cluster, addId = FALSE, addBorder = FALSE,   sort = TRUE)
```

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
