# Peer review of "Categorical Functional Data Analysis. The cfda R Package"

_mathematics, doi:10.3390/math9233074_

Round 1
Reviewer 1 Report
Review of "Categorical functional data analysis. The cfda R package" by Preda et al. (2021)
Authors presented the Categorical functional data analysis (CFDA) and its implementation on R package. Specifically, they considered the theoretical framework of Refs. [6] and [8], to implement CFDA in R package. The paper is well written and illustrations of application are clear. Thus the paper deserves to be published in Mathematics journal. I have some few comments/suggestions to be addressed by the authors:
1. First paragraph of page 2: put each reference in the right side of each application. For a application in economy, add the reference Idrovo-Aguirre et al. (2021), where Markov-Switching model is considered for data modelling.
2. Section 3: R package is also available at CRAN: https://cran.r-project.org/web/packages/cfda/index.html. Please mention this instead of GitHub availability.
3. Authors could mention that CFDA can be focused in outlier detection, such as in Dai et al. (2020), or interpreting the relationship between both methods.
References:
Dai, W., Mrkvicka, T., Sun, Y., Genton, M.G. (2020). Functional outlier detection and taxonomy by sequential transformations. Computational Statistics & Data Analysis, 149, 106960.
Idrovo-Aguirre, B.J., Lozano, F.J., Contreras-Reyes, J.E. (2021). Prosperity or Real Estate Bubble? Exuberance probability index of the Real Price of Housing in Chile. International Journal of Financial Studies 9(3), 51.
Author Response
Thank you for your report.
Please see in the attachment our answer.
Cristian Preda

Reviewer 2 Report
Paper considers an interesting and not sufficiently explored field of functional data analysis. Authors focus on aspect, they call, categorical functional data. I have an issue with this name.
- Authors require certain order of between points in the category space, so it should be considered ordinal not categorical.
- Moreover, I’m not sure if it should not to be extended to even discrete space, also I would be interested how theory would work in the case of spaces with infinite states (either countable or not).
I feel that the theory of so called principal components is a bit overwhelming the entire paper. First of all authors should expand on what would be the benefit of such decomposition. It cannot be memory reduction, as even single PC requires more than original function. Also the computational cost of such computation seems extensive. So please provide clear application. Also id like to see an example (graphical) of representing original function with PCs to see how they number factors in representativity.
For the package itself I regard it rather highly. All the statistics used are very useful and representation as a markov chain is very practical. I think that most of the functions will be used. The optimal encoding part however, is at least in the presented context just a firework with reduced usefulness.
I’d be happy if authors would discuss possibilities of porting their package into different languages, especially Julia and Python.
Author Response

(The authors gave the same response as above.)

Reviewer 3 Report
Report “Categorical functional data analysis. The cfda R package”:
The recent paper seems to be interesting and attract many authors to develop and cite it. The methodology of the paper is implemented in the cfda R package and is illustrated using a real data set in the clustering framework. I recommend the paper to be published in mathematics journal after the following minor comments:
- The values of Table 1 should be corrected from “e-4” to be “10^4”.
- The resolution of many plots needs to be improved.
- The title of Figure 13 (a) and (b) is overlapped.
- All R codes need to be collected in the appendix along with all its corresponding and related packages/libraires.
- Each equation should be followed by “.” or “,”.
- A list of all appreciations should be added in the appendix.
- Authors should illustrate how other readers expand this work. You may add some potential future works.
- In birth-and-death process, I think simulations should be re-performed for N=1000 not only for N=1000.
- In what way you choose the true values in simulation study. I think a certain methodology should be demonstrated.
- An extensive English revised should done.
Author Response

(The authors gave the same response as above.)
